# Vemurafenib Inhibits Acute and Chronic Enterovirus Infection by Affecting Cellular Kinase Phosphatidylinositol 4-Kinase Type III$\beta$

Mira Laajala,[a] Marleen Zwaagstra,[b] Mari Martikainen,[a] Magloire Pandoua Nekoua,[c] Mehdi Benkahla,[c] Famara Sane,[c] Emily Gervais,[d] Grace Campagnola,[d] Anni Honkimaa,[e] Amir-Babak Sioofy-Khojine,[e] Heikki Hyöty,[e,f] Ravi Ojha,[g] Marie Bailliot,[g] Giuseppe Balistreri,[g,h] Olve Peersen,[d] Didier Hober,[c] Frank Van Kuppeveld,[b] Varpu Marjomäki[a]

aDepartment of Biological and Environmental Science/Nanoscience Center, University of Jyväskylä, Jyväskylä, Finland

bSection of Virology, Division of Infectious Diseases & Immunology, Department of Biomolecular Health Sciences, Utrecht University, Utrecht, The Netherlands

cLaboratoire de Virologie ULR3610, Université de Lille, CHU Lille, Lille, France

dDepartment of Biochemistry & Molecular Biology, Colorado State University, Fort Collins, Colorado, USA

eDepartment of Virology, Tampere University, Faculty of Medicine and Health Technology, Tampere, Finland

fFimlab Laboratories, Tampere, Finland

gDepartment of Virology, Faculty of Medicine, University of Helsinki, Helsinki, Finland

hQueensland Brain Institute, The University of Queensland, Brisbane, Queensland, Australia

**ABSTRACT** Enteroviruses are one of the most abundant viruses causing mild to serious acute infections in humans and also contributing to chronic diseases like type 1 diabetes. Presently, there are no approved antiviral drugs against enteroviruses. Here, we studied the potency of vemurafenib, an FDA-approved RAF kinase inhibitor for treating BRAF$^{V600E}$ mutant-related melanoma, as an antiviral against enteroviruses. We showed that vemurafenib prevented enterovirus translation and replication at low micromolar dosage in an RAF/MEK/ERK-independent manner. Vemurafenib was effective against group A, B, and C enteroviruses, as well as rhinovirus, but not parechovirus or more remote viruses such as Semliki Forest virus, adenovirus, and respiratory syncytial virus. The inhibitory effect was related to a cellular phosphatidylinositol 4-kinase type III$\beta$ (PI4KB), which has been shown to be important in the formation of enteroviral replication organelles. Vemurafenib prevented infection efficiently in acute cell models, eradicated infection in a chronic cell model, and lowered virus amounts in pancreas and heart in an acute mouse model. Altogether, instead of acting through the RAF/MEK/ERK pathway, vemurafenib affects the cellular PI4KB and, hence, enterovirus replication, opening new possibilities to evaluate further the potential of vemurafenib as a repurposed drug in clinical care.

**IMPORTANCE** Despite the prevalence and medical threat of enteroviruses, presently, there are no antivirals against them. Here, we show that vemurafenib, an FDA-approved RAF kinase inhibitor for treating BRAF$^{V600E}$ mutant-related melanoma, prevents enterovirus translation and replication. Vemurafenib shows efficacy against group A, B, and C enteroviruses, as well as rhinovirus, but not parechovirus or more remote viruses such as Semliki Forest virus, adenovirus, and respiratory syncytial virus. The inhibitory effect acts through cellular phosphatidylinositol 4-kinase type III$\beta$ (PI4KB), which has been shown to be important in the formation of enteroviral replication organelles. Vemurafenib prevents infection efficiently in acute cell models, eradicates infection in a chronic cell model, and lowers virus amounts in pancreas and heart in an acute mouse model. Our findings open new possibilities to develop drugs against enteroviruses and give hope for repurposing vemurafenib as an antiviral drug against enteroviruses.

**KEYWORDS** acute infection, antiviral, chronic infection, drug repurposing, enterovirus

Address correspondence to Varpu Marjomäki, varpu.s.marjomaki@jyu.fi.

The authors declare a conflict of interest. H.Hy is a minor shareholder and member of the board of Vactech Ltd, which develops vaccines against picornaviruses. No other potential conflicts of interest relevant to this article were reported.

Enteroviruses are human pathogens that can cause infections ranging from mild symptoms, such as the common cold, to more severe diseases, such as aseptic meningitis or myocarditis (1). They are also known to contribute to the pathogenesis of some chronic diseases such as type 1 diabetes (2–4). Despite their prevalence and medical threat, there are no approved antiviral drugs against enteroviruses at the moment. In addition, vaccines are only available against poliovirus and enterovirus 71 (5). Due to the lack of antivirals and the fact that preparation of vaccines against all enterovirus serotypes is not feasible, there is a clear need for antivirals that can combat different enteroviruses.

Enteroviruses belong to the picornavirus family of nonenveloped viruses with a single-stranded positive-sense RNA genome enclosed by an icosahedral capsid. The capsid is composed of four different proteins, VP1 to VP4, and the genome also encodes nonstructural proteins 2A-2C and 3A-3D, which have different roles in viral replication, translation, and host cell manipulation to promote infection. In addition to viral proteins, many cellular factors are also important for efficient infection.

The replication of enteroviruses needs not only the viral 3D polymerase but also other viral nonstructural proteins that help to promote the remodeling of cellular membranes during replication into replication organelles (ROs) (6, 7). In addition, cellular factors are needed for efficient replication, including phosphatidylinositol 4-kinase type III$\beta$ (PI4KB), which is recruited by viral protein 3A/3AB and induces the formation of phosphatidylinositol 4-phosphate (PI4P)-enriched membranes in the replication sites (8–11). This has been shown to be important both for the formation of the ROs as well as replication (10).

Vemurafenib (PLX4032, RG7204) is an inhibitor of the V600E mutated form of BRAF and was approved by the FDA for the treatment of metastatic or unrespectable melanoma in 2011, based on clinical BRIM-3 studies (12). BRAF is a serine-threonine kinase of the RAF family that acts in the RAF/MEK/ERK signaling pathway, which normally regulates cell growth, survival, and differentiation. Mutations of the kinase lead to overactive signaling and hence excessive proliferation and survival of cancer cells. Vemurafenib inhibits BRAF$^{V600E}$ by directly binding to the kinase and blocking the activated pathway and downstream signaling (13).

In this study, we show that vemurafenib inhibits enterovirus infection in an RAF/MEK/ERK-independent manner. More specifically, we demonstrate that vemurafenib prevents enterovirus replication by affecting the cellular kinase PI4KB to lower PI4P lipid formation. In addition, we show that mutation in viral 3A can rescue the infection, indicating that the mutated virus is able to bypass the need for PI4KB kinase and PI4P lipids.

## RESULTS

**Vemurafenib inhibits the infection of enterovirus B and C group viruses and rhinovirus B.** In our preliminary drug screen comprising 462 different inhibitors against cellular kinases and metabolic actors, one drug, vemurafenib, was selected for further experiments due to its efficacy against echovirus 1 (EV1). We next tested whether vemurafenib has efficacy also for other enteroviruses. The efficacy of the drug was tested by evaluating the effect on virus-induced cytopathic effect (CPE). The results showed that in addition to EV1 infection, vemurafenib rescued the cells from coxsackie virus A9 (CVA9) infection in a dose-dependent manner (Fig. 1A, left). Similarly, also, all the coxsackievirus B (CVB) serotypes from 1 to 6 were inhibited by vemurafenib (Fig. 1A, right). We also evaluated the cytotoxicity of vemurafenib during both short (6.5 h) and long time points (24 h) in A549 cells (Fig. 1B). The results showed that the 50% cytotoxic concentration (CC$_{50}$) was 36 $\mu$M at the 24-h time point, while 75% of the cells were still viable after treatment with 100 $\mu$M vemurafenib for 6.5 h (Fig. 1B). Calculation of the 50% inhibitory concentration (IC$_{50}$) values and selectivity indexes (SIs) for different viruses showed that vemurafenib prevented the infection of EV1 most efficiently (Fig. 1A and B). In addition, poliovirus type 3 infection was efficiently rescued, and the cell viability reached control cell levels with low dosages of vemurafenib (Fig. 1C). In addition, we also wanted to test the efficacy of vemurafenib against rhinoviruses. HeLa cells were infected with human rhinovirus B14 (HRVB14) for 6 h with and without vemurafenib and immunolabeled

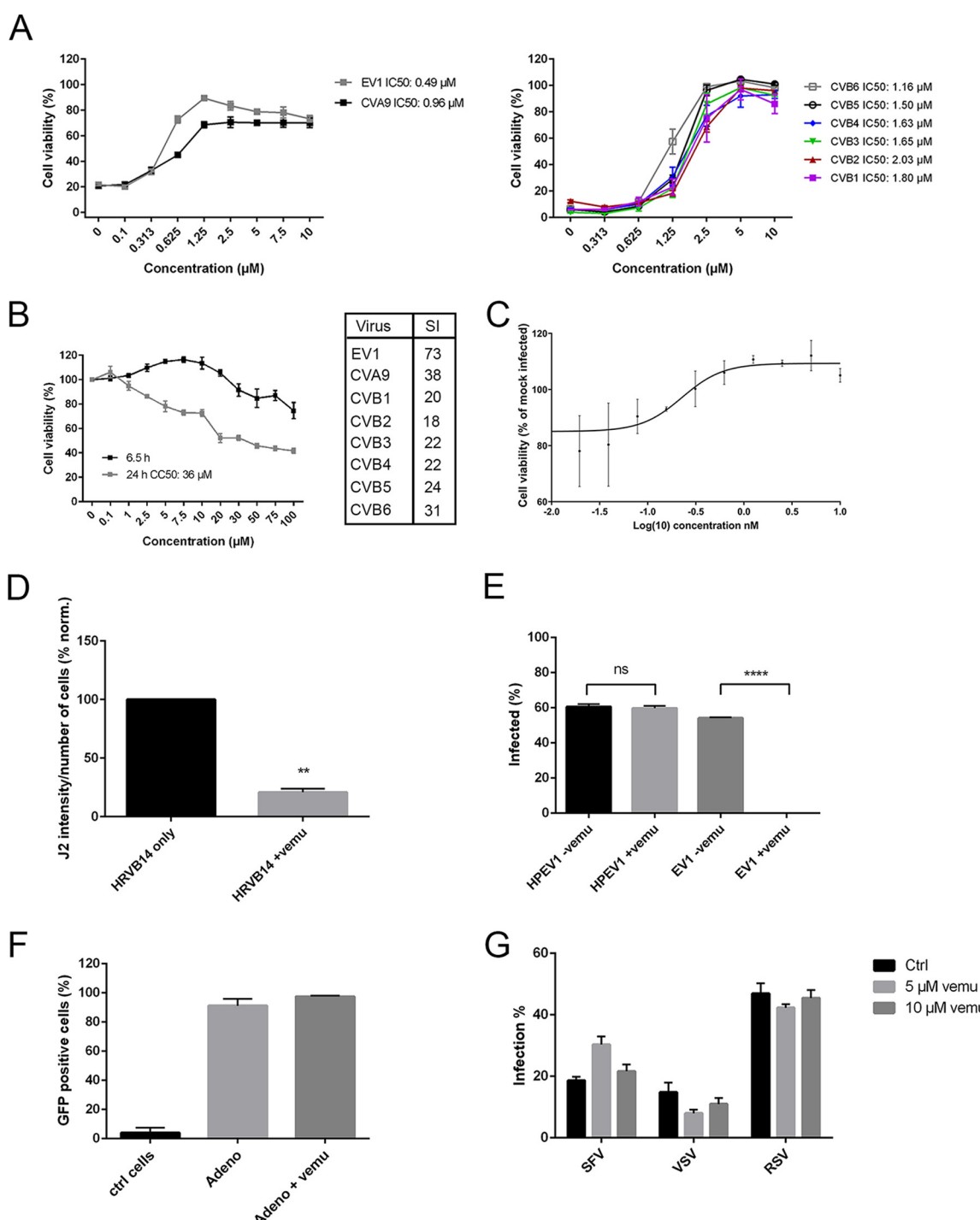

**FIG 1** Vemurafenib inhibits enterovirus infection in a dose-dependent manner. (A) A549 cells were treated with different concentrations of vemurafenib before infecting the cells with EV1 and CVA9 (left) or different CVB serotypes, 1 to 6 (right). After 18 h postinfection, the cell viability was determined by crystal violet staining. The cell viability was normalized to control sample with no virus infection and 0 $\mu$M concentration of vemurafenib. The results are mean from two repeats with three technical replicates $\pm$ SEM (left) or three replicates $\pm$ SD (right). $IC_{50}$ values were calculated using nonlinear regression analysis. (B) A549 cells were treated with different concentrations of vemurafenib, and the cytotoxicity was determined after short (6.5 h) and long time points (24 h) using CellTiter-Glo kit. The results are means from two repeats with three technical replicates $\pm$ SEM. $CC_{50}$ values were calculated using nonlinear regression analysis, and selectivity indexes (SIs) for different viruses were determined by $IC_{50}$ and $CC_{50}$. (C) Different concentrations of vemurafenib were tested against poliovirus type 3 by treating A549 cells with the drug for 1 h and then infecting the cells with the virus (MOI of 0.1). Cell viability was determined using CellTiter-Glo kit after 2 days of infection. The results are means $\pm$ SDs. (D) Vemurafenib decreases the replication of human rhinovirus B14 (HRVB14) indicated by labeling of dsRNA. HeLa cells were infected with HRVB14, and infection was allowed to proceed for 6 h at 34°C. After fixation with 4% PFA, the cells were immunolabeled with dsRNA antibody (J2) to determine the status of virus replication. The intensity of J2 signal was calculated using BioImageXD software. The J2 intensity was normalized to the total

against double-stranded RNA (dsRNA) to show the status of viral replication. The results showed that vemurafenib treatment decreased the replication of HRVB14 by 80%, indicating that, in addition to enterovirus species, rhinovirus infection is also inhibited by vemurafenib (Fig. 1D).

As we observed that vemurafenib efficiently inhibited the infection of different enteroviruses, we next wanted to study whether vemurafenib would act more broadly against different virus types. First, we tested a close relative of enteroviruses, namely, human parechovirus 1 (HPEV1), also belonging to the picornavirus family. Our immunofluorescence staining of viral capsid protein VP1 and imaging results showed that the number of HPEV1-infected cells was the same (around 60%) with and without vemurafenib treatment, indicating that the drug did not prevent the infection of HPEV1 (Fig. 1E). As a control, we used EV1 in the same assay and showed that the number of EV1-infected cells dropped drastically after vemurafenib treatment, in contrast to the high infection (around 55%) of control cells (Fig. 1E). Next, we studied the infection of a DNA virus, namely, adenovirus 5, using a green fluorescent protein (GFP) expression construct of the virus. The results were detected by flow cytometry and showed that in the presence of vemurafenib, the number of GFP-positive cells was the same as in control infection (around 90%), while autofluorescence from noninfected cells was low (Fig. 1F). Finally, we wanted to test the efficacy of vemurafenib on Semliki Forest virus (SFV), another single-stranded positive-sense RNA virus, as well as vesicular stomatitis virus (VSV) and respiratory syncytial virus (RSV) as two single-stranded negative-sense RNA viruses. Using GFP-expressing variants of the viruses, the infection percentage was determined in HeLa cells after 6 h postinfection (SFV and VSV) or 18 h postinfection (RSV). The GFP-positive cells were determined by imaging, and the results showed that with all three viruses, neither of the tested concentrations of vemurafenib (5 or 10 $\mu$M) reduced the number of infected cells (Fig. 1G). All in all, our results showed that vemurafenib has a selective effect on all tested enteroviruses, including rhinovirus, and does not extend to a broader variety of RNA and DNA viruses tested.

**Vemurafenib affects both acute and chronic CVB4 infection.** Next, we wanted to study whether vemurafenib can inhibit the infection of CVB4 also in pancreas cells, which are biologically relevant target cells when studying the secondary infections caused by enteroviruses. Our results showed that vemurafenib also efficiently rescued mouse pancreas cells from CVB4 infection, as the cell viability reached control level with a low micromolar concentration of vemurafenib (Fig. 2A). More importantly, the efficacy of vemurafenib was proven in a mouse model of acute CVB4 infection (Fig. 2B). During the 5-day experiment, the mice were treated daily with vemurafenib and infected on the second day with CVB4. The results showed that the levels of infectious CVB4 particles decreased by 2 to 3 log in pancreas and heart tissue of mice when the mice were treated with vemurafenib (Fig. 2B).

Next, we tested whether vemurafenib can eradicate a persistent CVB4 infection in human pancreas cells. We were excited to find out that in this chronic infection model, the viral titers were lowered below the detection limit ($10^{1.5}$ 50% tissue culture infectious dose/mL [$TCID_{50}$/mL]) within 10 days of infection (Fig. 2C). In comparison, the viral titers increased during the whole 30 days of infection up to $10^{7.5}$ $TCID_{50}$/mL in the

**FIG 1** Legend (Continued)

number of cells determined by DAPI staining. Finally, the results were normalized to HRVB14 infection without the drug, which was set to 100%. Values are means from two separate experiments $\pm$ SEM. Unpaired $t$ test with Welch's correction was carried out after arcsine transformation. **, $P < 0.01$. (E) Vemurafenib does not inhibit infection of human parechovirus 1 (HPEV1) belonging to the *Picornaviridae* family. The drug was administered in A549 cells, and it was present for the whole duration of the infection (6 h) of purified HPEV1 or EV1. The cells show high infectivity, which was detected by the accumulation of capsid protein VP1 in the cell cytoplasm revealed by immunofluorescent labeling. Infection percentage was calculated by comparing VP1-positive cells to the total cell amount calculated based on DAPI staining. The results are means $\pm$SEMs from two different experiments. Unpaired $t$ test with Welch's correction was carried out after arcsine transformation. ns, not significant; ****, $P < 0.0001$. (F) Vemurafenib is not effective against adenoviruses. The effect on adenovirus 5 transduction was tested by using a GFP expression construct after an overnight exposure on A549 cells. The GFP fluorescence, proof of successful transduction, was similarly effective in the presence of the drug. The positive cells out of all cells were counted. Also, control cells were included to detect the autofluorescence in the cells. The results show mean values from two experiments $\pm$ SEMs. (G) Vemurafenib does not prevent the infection of Semliki Forest virus (SFV), vesicular stomatitis virus (VSV), and respiratory syncytial virus (RSV). HeLa cells were pretreated with 5 or 10 $\mu$M vemurafenib or DMSO as control for 30 min, after which GFP-expressing variants of the viruses were added. The infection was allowed to proceed for 6 h (SFV and VSV) or 18 h (RSV), and the infection percentage was determined based on GFP fluorescence. The values are mean of three independent experiments $\pm$ SD.

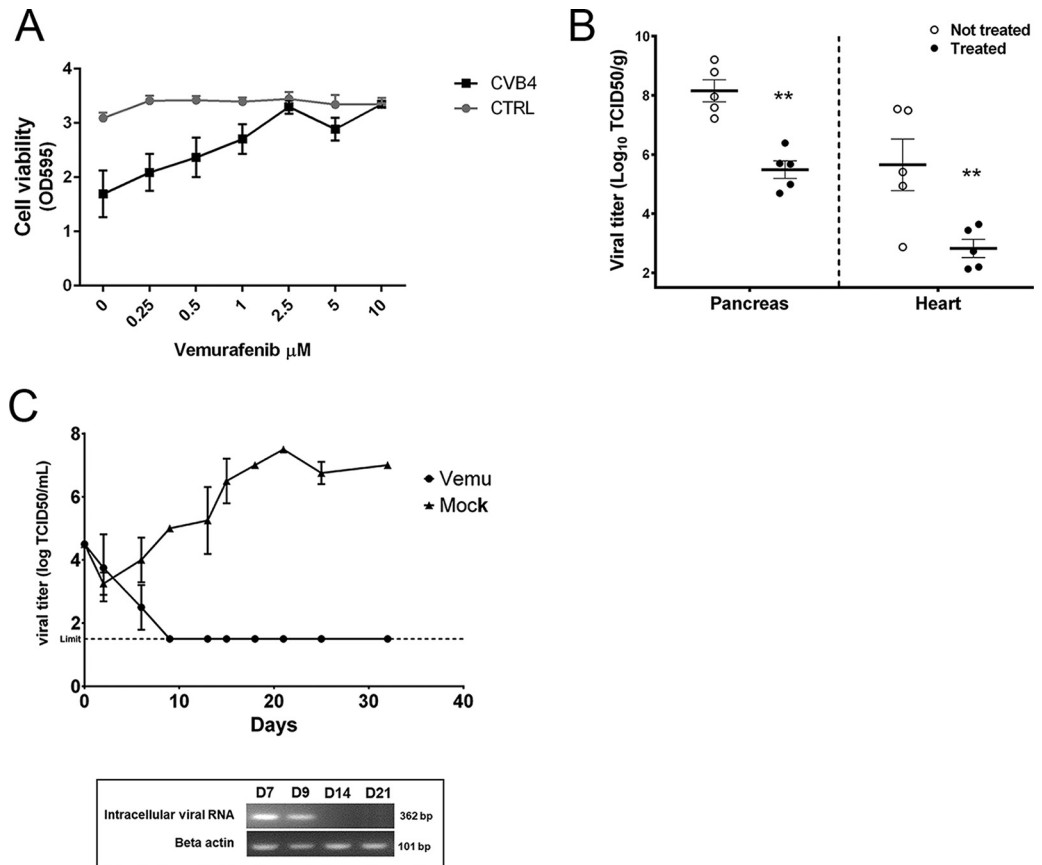

**FIG 2** Vemurafenib affects both acute and chronic CVB4 infection. (A) Vemurafenib prevents the infection of CVB4 in pancreatic beta cell line in a dose-dependent manner. Mouse pancreatic beta cells, Min-6 cells, were treated with different concentrations of vemurafenib and infected or not with CVB4. The viability of cells was determined using crystal violet staining. The values are means ± SEMs from three replicates. (B) Vemurafenib decreases the levels of infectious CVB4 viral particles in the heart and pancreas tissue of mice. Hsd:ICR (CD-1) female mice at the age of 3 weeks were inoculated intraperitoneally with vemurafenib dissolved in DMSO and diluted in PBS (10 mg/kg) or with DMSO diluted in PBS once a day (starting on day 1) for 5 days. The animals were inoculated intraperitoneally with CV-B4 E2 ($6 \times 10^6$ $TCID_{50}$ in 200 $\mu$L PBS) on day 2. The animals were sacrificed on day 6, blood was collected, and portions of each organ (pancreas and heart) were frozen for determination of viral titer. Frozen organs were weighed, crushed using a TissueRuptor, homogenized in 0.5 mL of PBS, and then centrifuged at $2,000 \times g$ for 10 min at 4°C. The supernatants were harvested to measure the titer of infectious particles (on HEp-2 cells), and titers were normalized to tissue weight. The results were expressed as log $TCID_{50}$ per gram. The limit of detection of the test was 0.75 log $TCID_{50}$/g. (C) Vemurafenib eradicates persistent CVB4 infection. The level of infectious viral particles in persistently infected human Panc-1 cells was determined by quantifying the viral titers in the supernatant of treated (circles) and untreated cells (triangles). The values are means ± SDs from two experiments. In addition, the presence of viral RNA was studied using RT-PCR and seminested PCR, followed by electrophoresis of the amplification products in agarose gel at days 7, 9, 14, and 21.

mock-treated cells. In addition to the viral titers, the levels of intracellular viral RNA were also studied on days 7, 9, 14, and 21 using reverse transcriptase PCR (RT-PCR) and seminested PCR. The results showed that in vemurafenib-treated cells, the viral RNA was detectable on days 7 and 9 but not anymore on days 14 and 21, indicating that the drug had eradicated the infection (Fig. 2C).

**The inhibitory effect of vemurafenib is not through BRAF signaling.** Since the cellular target of vemurafenib is the $BRAF^{V600E}$, one of the potential mechanisms for the virus inhibition could be through this BRAF kinase. On the other hand, our results demonstrated that another BRAF inhibitor, sorafenib, did not inhibit the infection of EV1 and CVB3 (Fig. 3A). More importantly, we showed the inhibitory effect of vemurafenib in A549 cells, which do not carry the mutated $BRAF^{V600E}$ (14), further suggesting that enterovirus infection is not halted due to the inhibition of BRAF, but some other mechanism. In addition, the inhibition of Ras, which is upstream from BRAF, had only a mild inhibitory effect on the infection of EV1 and CVB3 when a Ras inhibitor, Salirasib,

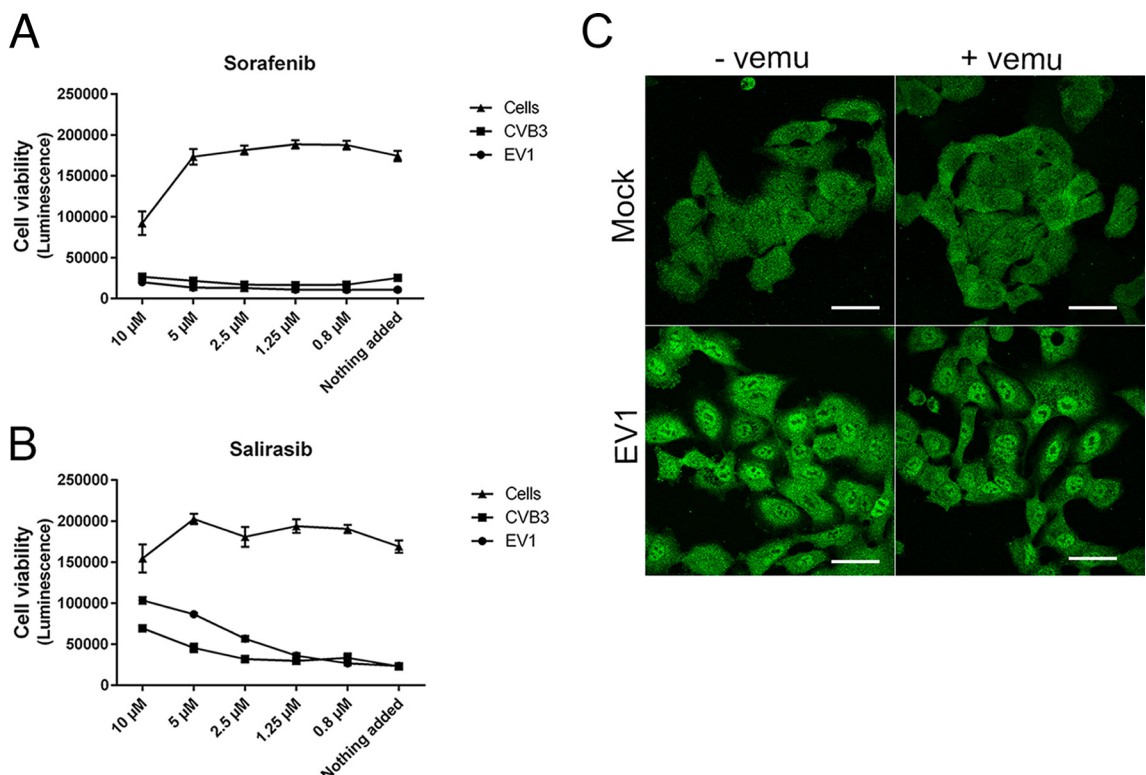

**FIG 3** The inhibitory mechanism of vemurafenib is not through BRAF signaling. A549 cells were treated with different concentrations of a BRAF inhibitor, sorafenib (A), or Ras inhibitor, Salirasib (B), before EV1 or CVB3 infection. Also, noninfected control cells were included. After 18 h postinfection, the cell viability was determined by CTG assay. Values are means $\pm$ SEM from three replicates. (C) Confocal images of A549 cells infected with EV1 for 2 h with or without vemurafenib treatment compared to mock-infected cells. ERK is immunolabeled green. Scale bar, 40 $\mu$M.

was tested (Fig. 3B). We also evaluated the effect of vemurafenib downstream from BRAF by studying ERK signaling during EV1 infection with or without vemurafenib. The results showed that ERK was activated during EV1 infection in A549 cells after 2 hour postinfection, which was seen as ERK translocation into the nucleus, and that vemurafenib did not affect this activation (Fig. 3C). All of these data thus indicate that the inhibitory effect of vemurafenib is not due to the inhibition of BRAF signaling, but instead, the virus infection is inhibited by other mechanisms.

**Vemurafenib does not inhibit uncoating but causes an arrest early during infection after RNA release from endosomes.** Since vemurafenib did not seem to inhibit enterovirus infection through BRAF, we wanted to study in detail, step by step, at what stage vemurafenib exerts its effect against enteroviruses. We first wanted to study the direct effects on the virus particle itself by using a real-time spectroscopic assay (Fig. 4A). In the assay, we can monitor the stability of the virus and see in real time if the virus switches to the intermediate uncoating form (IMF) and then finally to the empty form, totally releasing the RNA outside from the virion (15–17). The IMF shows high fluorescence upon SYBR green II addition due to the permeability of the IMF to small molecules such as SYBR green II intercalating to the RNA. The empty form also shows high fluorescence, but it can be killed by adding RNase to the assay. We can thus faithfully monitor the presence of three different forms of the virus. The results showed that in storage buffer conditions (2 mM MgCl$_2$ in phosphate-buffered saline [PBS]), the particle was stable for 3 h at 37°C, and hence, the fluorescence did not increase with or without vemurafenib being present (Fig. 4A). On the other hand, in ionic conditions (20 mM NaCl, 6 mM KH$_2$PO$_4$, 12 mM K$_2$HPO$_4$, and 0.01% fatty acid-free bovine serum albumin [faf-BSA]), which we have earlier shown to open the viral particle (17), the viral RNA was released from the capsid, but again in the same manner with or without vemurafenib being present or not (Fig. 4A). In addition, we carried out

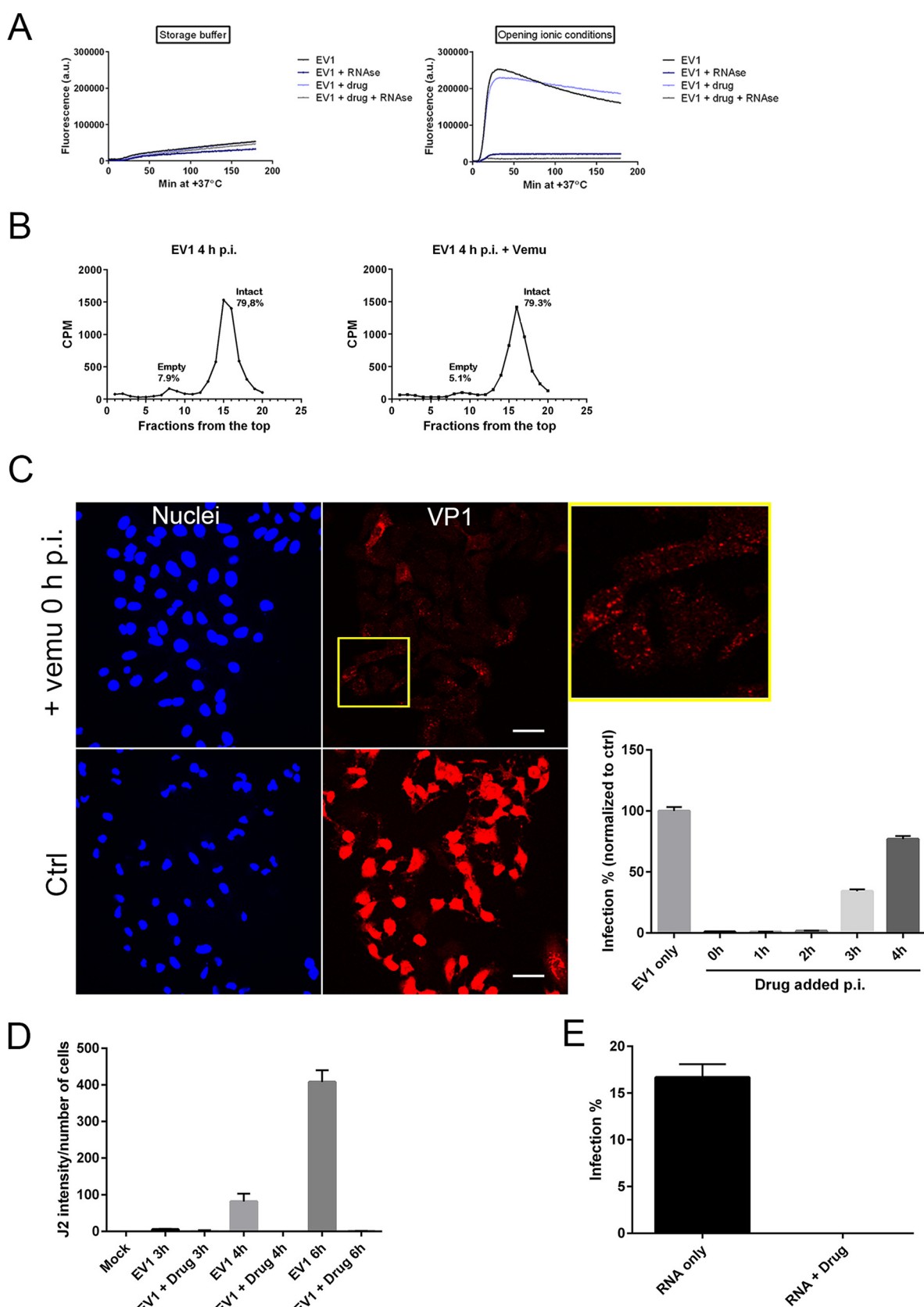

**FIG 4** Vemurafenib does not affect uncoating but prevents the translation and replication of EV1. (A) Direct effects of vemurafenib on EV1 particles was studied using a real-time spectroscopic assay, where the viral opening was monitored using SYBR green II, which

an uncoating assay using radioactive virus in a sucrose gradient, which showed that after 4 h infection in cells, the virus uncoats in a similar proportion whether vemurafenib is present or not, which was detected as the number of empty particles (Fig. 4B). From these data, we can conclude that vemurafenib does not break down or stabilize the virus particle directly, and instead, the virus seems to uncoat normally.

To identify the timing when vemurafenib inhibits EV1 infection, we performed a time-of-addition experiment, where vemurafenib was added at different times postinfection. We found that EV1 infection was inhibited most efficiently if vemurafenib was added 2 to 3 hours postinfection (hpi) or before (Fig. 4C). The infection was evaluated by immunolabeling the VP1 capsid protein of EV1, which was detected as high levels in the cytoplasm of the control cells, but only showed the input virus in endosomes in vemurafenib-treated cells (Fig. 4C), indicating that the protein production of EV1 was halted. In addition to EV1, protein production was halted during EV71 infection, as only the input virus was visible in the endosomes (see Fig. S1 in the supplemental material). We also measured EV1 replication in cells by labeling the dsRNA in cells, which appeared soon after the onset of replication. This labeling and quantification of signal from cells showed that the replication of EV1 was totally inhibited by vemurafenib, whereas in control infection, the intensity of the dsRNA label started increasing after 3 to 4 hpi (Fig. 4D). More importantly, the infection was also inhibited if the endosomal step was bypassed completely and the viral RNA was directly transfected into the cells (Fig. 4E), indicating that vemurafenib affects the step(s) after RNA release from the endosomes into the cytoplasm.

**Vemurafenib has no effect on viral proteases or 3D polymerase.** As we showed that vemurafenib caused an arrest in infection if added before the translation and replication were fully ongoing, next, we wanted to study the steps after RNA release in more detail. We first addressed the cleavage of the polyprotein into single proteins, which is carried out by the viral 2A and 3C proteases. Hence, we wanted to study whether vemurafenib inhibits the viral proteases directly. In addition to the viral polyprotein, the viral proteases also have cellular targets which they cleave to promote the infection. We studied the action of the proteases by detecting the cleavage of the known targets of 2A and 3C, poly(A) binding protein (PABP) and stress granule assembly factor 1 (G3BP1), respectively. A549 cell homogenate was incubated with the CVB3 proteases *in vitro*, and the cleavage of PABP and G3BP1 was detected using Western blotting. Our results showed that 2A protease cleaved PABP even if increasing amounts of vemurafenib were added (Fig. 5A). In addition, the cleavage action of 3C was not affected by vemurafenib, as the cleavage product of G3BP1 appeared both with and without vemurafenib treatment (Fig. 5A). Next, we wanted to study whether vemurafenib inhibits the viral 3D polymerase, which is the key protein in viral replication. Polymerase initiation and elongation assays were used to determine if vemurafenib had inhibitory effects on $3D^{pol}$-RNA complex formation, incorporation of the first nucleoside triphosphates (NTPs), or processive elongation. Two fluorescently labeled hairpin primer/template RNAs were used to track the rapid elongation of prebound RNA1 complexes and the slow binding followed by

**FIG 4** Legend (Continued)

intercalates with viral RNA, causing an increase in fluorescence. The opening was studied in storage buffer (2 mM $MgCl_2$ in PBS) and opening ionic conditions (20 mM NaCl, 6 mM $KH_2PO_4$, 12 mM $K_2HPO_4$, and 0.01% faf-BSA). RNase was added to the reactions in order to separate the fluorescence that comes from the inside the particle (porous particle) from the fluorescence of the RNA outside the capsid. (B) Radioactively labeled EV1 particles were used to infect A549 cells. After 4 hpi, the cell lysates were treated with detergent, collected, and run in 5 to 20% sucrose gradients. The proportional amount of empty and intact particles was calculated by dividing the counts per minute of incident fractions with the total counts per minute value. The gradients are representative results of 3 three different experiments. (C) A549 cells were infected with EV1, after which vemurafenib was added at different times postinfection. The infection was determined by immunolabeling the viral capsid protein VP1 (red) and comparing the number of VP1-positive cells to the total cell number calculated based on DAPI staining (blue). The results were normalized to control infection where no drug was added. Values are means ± SEM from three replicates. Scale bar, 40 $\mu$M. (D) EV1 replication in A549 cells was measured in the presence or absence of vemurafenib at 3, 4, or 6 hpi. The dsRNA structures were immunolabeled with J2 antibody, and the intensity of the signal was calculated using BioImageXD software. A control with no virus was also included (mock) to determine the background of the antibody. The J2 intensity was normalized to the total number of cells determined by DAPI staining. Values are means ± SEMs from two replicates. (E) EV1 RNA was transfected into A549 cells with or without vemurafenib, and the infection was determined after 6 hpi by immunolabeling EV1 capsid protein VP1. The infection percentage was calculated by comparing the number of VP1-positive cells to the total cell number calculated based on DAPI staining. Values are means ± SEM from two replicates.

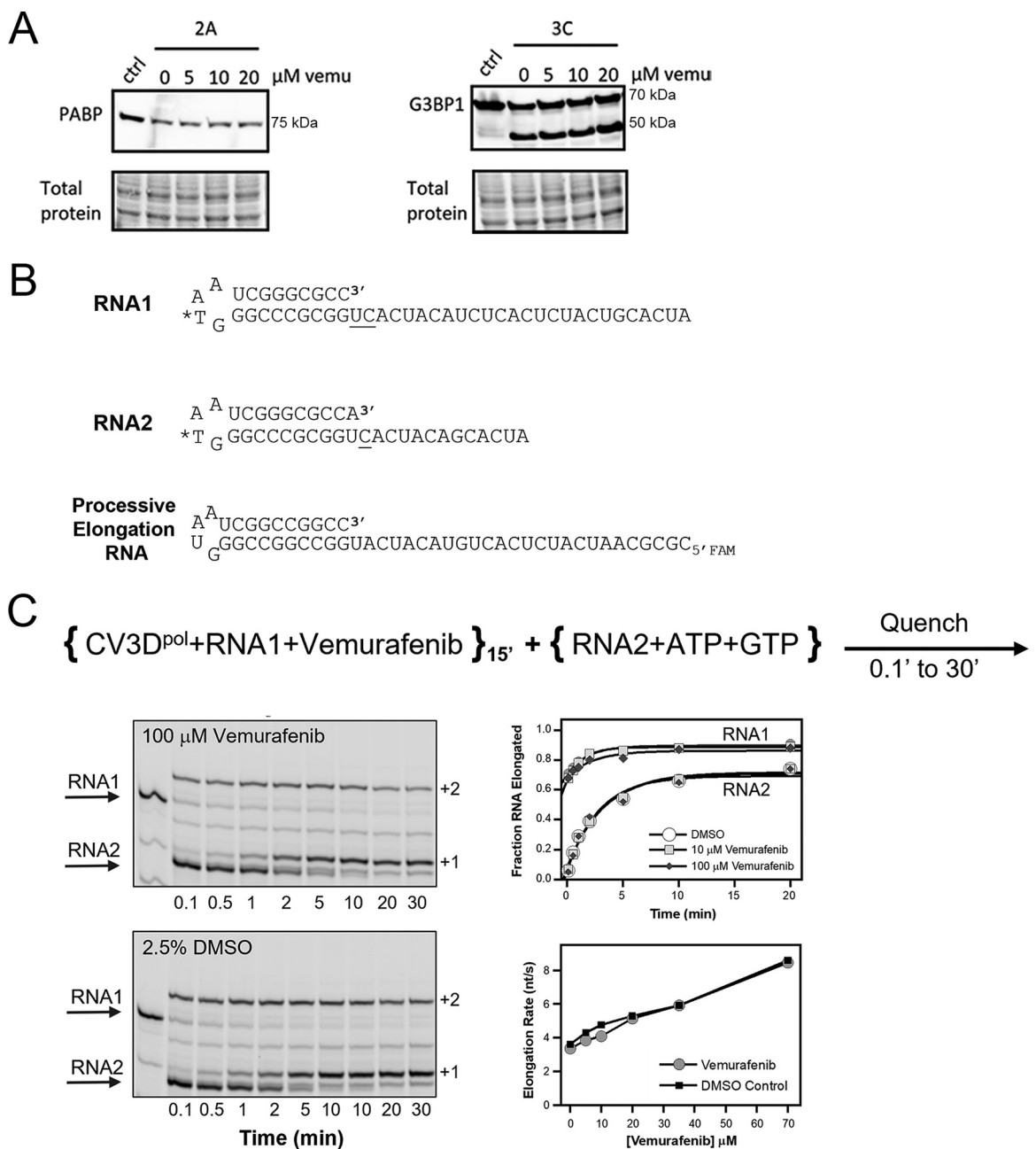

**FIG 5** Vemurafenib does not prevent the actions of 2A and 3C proteases and 3D polymerase. (A) The effect of vemurafenib on viral proteases was studied by treating A549 cell homogenate with purified 2A and 3C proteases with or without the drug (5, 10, or 20 μM) being present. The results were detected by Western blotting, and known cellular targets for 2A and 3C, poly(A) binding protein (PABP), and stress granule assembly factor 1 (G3BP1), respectively, were immunolabeled. The results are representative of two separate experiments. (B) RNAs used in assays to study the effect of vemurafenib on viral 3D polymerase function. Asterisk indicates IRDye800 labeling site. (C) CVB3 3D$^{pol}$ was incubated at room temperature for 15 min with RNA1 and various inhibitor concentrations or with 2.5% DMSO as a control. At 15 min, GTP and ATP were added along with RNA2, allowing the assembled 3D$^{pol}$-RNA1 complexes (≈60% of RNA1) to rapidly incorporate ATP and GTP and produce a +2 product, while RNA2 undergoes a slower reaction involving RNA binding followed by GTP incorporation to produce its +1 product. Top plot shows +2 and +1 product formation time courses for RNA1 and RNA2, respectively, and the bottom plot shows that vemurafenib has no effect on processive elongation rates measured by stopped-flow kinetics compared to concentration-matched DMSO controls (22.5°C, pH 6.5).

NTP incorporation with RNA2 (Fig. 5B). Vemurafenib did not inhibit RNA binding or elongation when preincubated for 15 min with CVB3 3D$^{pol}$ and RNA1 as seen by the rapid formation of a +2 RNA product upon addition of ATP and GTP. Similarly, vemurafenib did not inhibit the binding of 3D$^{pol}$ to RNA2 or the formation rate of the +1 RNA2 product compared to dimethyl sulfoxide (DMSO) controls (Fig. 5C). Rapid stopped-flow assays

were used to assess vemurafenib effects on processive elongation rate over an extended single-stranded RNA (ssRNA) template containing a 5′ fluorescein label (Fig. 5C). The polymerase rate is measured by a lag phase that occurs prior to a fluorescein fluorescence increase as the polymerase reaches the end of the RNA template (18). Elongation complexes mixed with increasing concentrations of vemurafenib showed no reduction in elongation rates, and in fact, elongation rates increased slightly because of the increasing DMSO concentrations associated with increasing amounts of vemurafenib added to the reactions (Fig. 5C). All together, these data indicate vemurafenib does not have significant effects on CVB3 polymerase elongation complex formation, initiation, or processive elongation.

**Vemurafenib affects PI4KB kinase and PI4P levels, which are important for enterovirus replication.** In addition to the viral polymerase, the replication of enteroviruses requires different cellular factors and other viral proteins. Remodeling of the cellular membranes into replication organelles is an important step during replication, and the key players of the event are, among others, viral 3A, cellular kinase PI4KB, and the PI4P lipids produced by the kinase. As we showed that the viral polymerase was not inhibited by vemurafenib, we next wanted to study the other important players during the replication step. We showed in noninfected cells that vemurafenib treatment changed the localization and expression of PI4KB (Fig. 6A), as has been shown before with PI4KB inhibitors (19). More importantly, also, the levels of PI4P lipids decreased drastically following vemurafenib treatment, which was detected using a plasmid encoding the PI4P sensor, a GFP-tagged pleckstrin-homology (PH) domain of four-phosphate adaptor protein 1 (FAPP1). As a positive control, we used a known PI4KB inhibitor, namely, BF738735, which showed similar results to vemurafenib treatment (Fig. 6A). In addition to PI4KB, we also wanted to study another important host factor related to viral replication, namely, cellular oxysterol binding protein (OSBP), which increases the cholesterol content in the replication membranes by mediating the exchange of PI4P lipids with cholesterol. As positive controls, we used two known OSBP inhibitors, OSW-1 and itraconazole, which caused a clear change in the localization of OSBP (Fig. 6B), as has been demonstrated before with OSBP ligand 25-hydroxycholesterol as well as OSBP inhibitors (20–22). In contrast, OSBP localization in vemurafenib-treated cells appeared similar to the control and BF738735-treated cells, indicating that vemurafenib (and BF738735) did not affect OSBP (Fig. 6B). Furthermore, using a *Renilla* luciferase construct, we were able to show that an H57Y mutation in viral 3A was able to protect against the effect of vemurafenib, as the $IC_{50}$ value was increased by 5-fold compared to the control virus (Fig. 6C). In contrast, an AVIVAV mutation in viral 2C protein showed more a similar $IC_{50}$ value (9.81 $\mu$M) to control infection (5.52 $\mu$M) (Fig. 6C). As controls, we used BF738735 and S-fluoxetine (SFX), whose $IC_{50}$ increased clearly during 3A-H57Y and 2C-AVIVAV infection, respectively (Fig. S2). All in all, our results suggest that vemurafenib has an effect on cellular PI4KB kinase, and a mutation in viral 3A can rescue the infection in the presence of vemurafenib.

## DISCUSSION

Enteroviruses are common human viruses that can cause a huge number of both mild, as well as more severe, diseases. Although many inhibitors have been found against enteroviruses, none of them have been approved on the market. For example, the 3C protease inhibitor rupintrivir (AG7088) and its orally bioavailable form, compound 1, were shown to inhibit many HRV serotypes, clinical isolates, and enteroviruses (23–26). The phase II clinical studies also showed that rupintrivir was able to prevent experimentally induced rhinovirus colds (27). However, further clinical development of rupintrivir was halted when it did not significantly have an effect on HRV infections in natural infection studies. The absence of antivirals against enterovirus infections and the difficulty of getting new inhibitors on the market have driven researchers to search novel strategies to fight against these viruses. The repurposing of drugs that are already well characterized and tested clinically and that are already on the market for other purposes is one of the more efficient strategies to find antivirals and put them more quickly into practice.

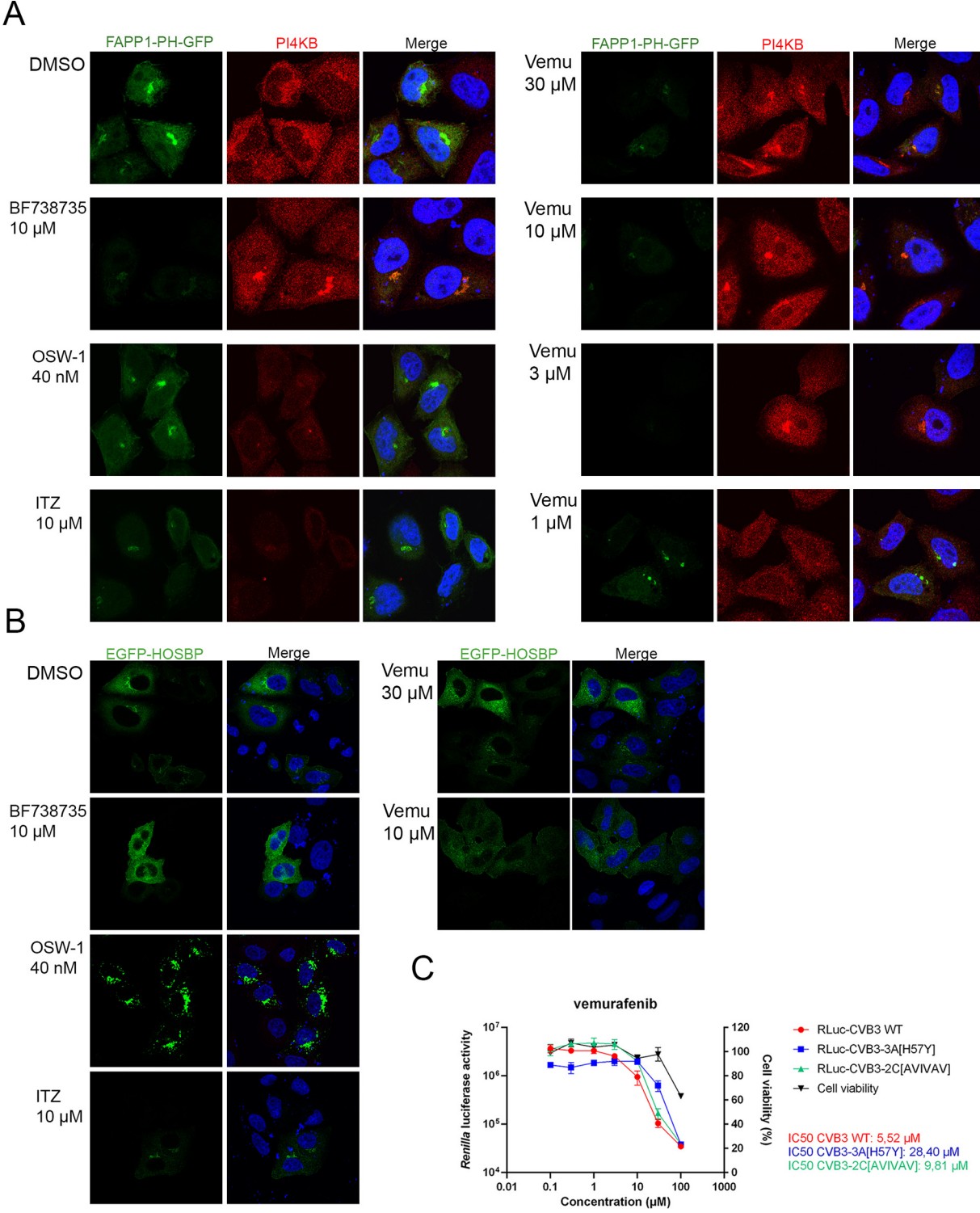

**FIG 6** Vemurafenib affects PI4KB kinase and PI4P lipids. After overnight transfection with FAPP1-PH-GFP plasmid (A) or EGFP-HOSBP plasmid (B), HeLa cells were treated with vemurafenib (vemu), BF738735, OSW-1, or itraconazole (ITZ) for 1 h. Subsequently, the cells were fixed and stained with an antibody against PI4KIII$\beta$ (PI4KB, red). Cell nuclei are visible in blue. (C) HeLa cells were infected with *Renilla* luciferase constructs RLuc-CVB3 WT, RLuc-CVB3-2C-AVIVAV, or RLuc-CVB3-3A-H57Y for 30 min (MOI of 0.1). Subsequently, the supernatant was replaced with DMEM containing the compound. Finally, the cells were lysed, and luciferase activity was measured or cell viability was determined at 7 hpi. Values are means ±SDs from three replicates, and IC$_{50}$ values were calculated using nonlinear regression analysis. The change in the IC$_{50}$ of H57Y mutant was statistically significant ($P = 0.0001$), while the change with AVIVAV mutant was not statistically significant ($P = 0.09$).

With this in mind, we set out to screen drugs that were already approved or under investigation in order to find inhibitors against enterovirus infections. The most potent hit, vemurafenib, which is already an FDA-approved drug and designed to treat melanoma patients, was studied further to find out the mechanism for the inhibition of virus infection. Interestingly, the antiviral effect of BRAF[V600E] inhibitor vemurafenib was not related to the inhibition of the BRAF kinase since the virus infection was efficiently inhibited in a cell line (A549) that does not contain the V600E mutation of BRAF (14). Furthermore, we showed that downstream signaling from BRAF to ERK was not affected by vemurafenib, as the enterovirus infection caused similar activation of ERK with or without vemurafenib treatment. In fact, it has been earlier shown that, paradoxically, vemurafenib activates RAF/MEK/ERK in BRAF[WT] cells (14, 28). Interestingly, it was also shown earlier that influenza A virus infection was inhibited by vemurafenib (29). In that study, the authors also showed that the RAF/MEK/ERK signaling cascade was activated by vemurafenib, but some other MAP kinases, like JNK and p38, were inhibited, which were suggested to impair viral protein expression and the virus-induced apoptosis.

Our mechanistic studies showed that the viral uncoating is not prevented and the inhibitory effect is prevented after uncoating, at 2 to 3 hpi, which is in the time frame of RNA release and early replication according to our earlier results. In a previous study, electron tomography and a fluorescence assay showed openings and a permeability increase in the multivesicular structures from where the RNA is released after 2 to 3 hpi, respectively (30). In addition, we have previously shown that EV1 replication starts around 3 hpi and then exponentially increases after 4 hpi, detected by the appearance of negative-sense RNA and positive-sense RNA by quantitative PCR (qPCR) (31). The start of replication was also shown in this study when dsRNA structures of EV1 slightly appeared at 3 hpi and were more evident at 4 hpi. We also showed that in the presence of vemurafenib, this replication was completely prevented.

When administered 2 hpi at the latest, vemurafenib efficiently inhibited translation and caused the accumulation of the input virus in endosomes, in contrast to the control infection, where the whole cytoplasm was full of newly formed capsid proteins. However, we did not know whether the block was in the translation itself, replication, or some steps in between. Detailed *in vitro* studies with purified viral proteases 2A and 3C demonstrated that their function was not impaired. Furthermore, 3D viral polymerase was able to elongate the RNA in a similar manner with or without vemurafenib treatment, showing that 3D polymerase function was not compromised either. Interestingly, while this article was in preparation, another group found that EV71 infection was also prevented by vemurafenib when the compound was added 2 hpi or earlier (32). More specifically, they showed that the virus replication and assembly were halted during vemurafenib treatment and that the RAF/MEK/ERK pathway was activated. However, the exact mechanism of how vemurafenib prevented the infection was not shown.

In addition to viral proteins, there are many essential cellular factors that are needed for efficient enterovirus infection. During replication, enteroviruses modulate the cellular membranes and hijack host proteins to form unique membranous structures, ROs (6, 7) One of the important host factors is PI4KB, which is recruited by viral 3A/3AB protein to ROs, resulting in increased amounts of PI4P lipids (8–11, 33). Inhibitors against PI4KB have been shown to efficiently prevent enterovirus and rhinovirus infections (34, 35). Here, we showed that vemurafenib also affected PI4KB, which resulted in a decrease of PI4P lipids, and understandably, that would block replication. Importantly, we showed that HPEV1 infection was not inhibited, which may be explained by the fact that although parechoviruses are close relatives to enteroviruses, they have many differences in their infection cycle, including in their RO formation (36, 37). However, the exact mechanism of how vemurafenib affects PI4KB needs further examination, as it is still unknown whether the drug targets PI4KB directly or through another cellular player. Unfortunately, we had no access to active PI4KB reagent to be able to show a possible direct effect on the kinase. However, using a knockout cell line, acyl-coenzyme A binding domain-containing 3 (ACBD3), has been shown to be critical in PI4KB recruitment by 3A (38). In addition,

guanine nucleotide exchange factor 1/guanine nucleotide exchange factor of ADP ribosylation factor (GBF1/Arf1) is regulated by 3A, resulting in PI4KB accumulation during RO formation (10, 39), indicating the complexity of the interacting players and, hence, also possible antiviral targets of vemurafenib. However, at least for some other PI4KB inhibitors (enviroxime, GW5074, and PIK93), it was demonstrated that they did not change the localization of GBF1/Arf1 (19). In addition, although a functional PI4KB/OSBP pathway has been shown earlier (22, 35, 40, 41), we ruled out OSBP as a target because vemurafenib did not cause any change in the localization of OSBP in comparison to two known OSBP inhibitors. OSBP is a regulator of lipid homeostasis and has been shown to exchange cholesterol with PI4P lipids in ROs and affect enterovirus infection (20, 42).

Host factors are potentially very good targets for antiviral development because of their high barrier to resistance. However, cytotoxicity and tolerance are important aspects that need to be carefully studied and considered when developing new drugs. Some earlier studies have brought up concerns related to PI4KB inhibitors, as they have prevented bone marrow and lymphocyte proliferation *in vitro* and caused lethality in mice (34, 43). However, another study showed no signs of toxicity in mice that were treated for 3 days with 25 mg/kg of the PI4KB inhibitor compound 2 (33). Also, here, vemurafenib did not cause toxicity when the mice were treated for 5 days with a 10-mg/kg dosage. In addition, the pharmacokinetic studies with vemurafenib have been carried out earlier, as it already is approved for treatment of melanoma. Generally, vemurafenib is well tolerated, and the most common side effects include fatigue, arthralgia, and dermatitis (44, 45). However, kidney injuries have been reported (46) and suggested to be caused by an off-target effect of ferrochelatase inhibition (47). The dosage of vemurafenib in the nephrotoxicity mouse model studied by Bai et al. (47) was higher (20 mg/kg, twice a day) than our virus studies (10 mg/kg, once a day), which may suggest that low dosages, which are sufficient for virus clearance, would not cause nephrotoxicity. Vemurafenib has also been successfully used with child cancer patients (48–51). It has to be kept in mind, though, that during treatment of life-threatening cancer, manageable toxicity is allowed, whereas more consideration of toxicity has to be taken during treatment of viral infections. However, treatment of cancer is usually a long process with months of drug therapy, while viral infection could be treated within days rather than months, which could decrease adverse side effects. In addition, lower dosages of vemurafenib could be sufficient for treating virus infections. Yet more studies are obviously needed to ensure efficient but safe dosage and treatment time of vemurafenib when treating enterovirus infection.

Although targeting host factors for antiviral therapy is generally a good strategy in terms of viral resistance, it has also been shown that enteroviruses can gain resistance against PI4KB and OSBP inhibitors (19, 41). By using a well-defined virus mutant shown to be resistant against PI4KB inhibitors (19), we showed that the CVB3 3A-H57Y mutant was able to infect in the presence of vemurafenib. However, when we repeatedly tried to develop escape mutants in the presence of vemurafenib using a protocol described by others (52–54), resistant mutants did not develop (data not shown). Interestingly, Arita et al. reported that another RAF inhibitor, GW5074, was able to prevent enterovirus infection in an RAF/MEK/ERK-independent manner through PI4KB and, more importantly, that the virus did not gain resistance to the drug after multiple passages (35, 55). It may well be that escape mutants do not develop as easily against vemurafenib and other RAF inhibitors targeting PI4KB, which could increase their potential as antivirals against enteroviruses.

Understanding the key host factors in the replication of different positive RNA viruses is essential for antiviral development. In general, all positive RNA viruses are dependent on the cellular membranes and lipid metabolism of the host, but the viruses can still be divided into two groups in terms of the morphology of their replication organelles (6). The invagination type of ROs has negative membrane curvature and is used by flavi- and alpha-like viruses, while vesicular-tubular ROs have positive-membrane curvature and are used by picornaviruses. In addition to the different RO morphologies, the host factors involved may differ within and between virus families. Here, we showed that another positive RNA virus, the alphavirus SFV, was not affected by vemurafenib, and in addition,

HPEV1 replication was not inhibited. Although HPEV1 belongs to picornaviruses, the ROs seem to differ from enteroviruses, and many enterovirus inhibitors have been shown to lack efficacy against parechoviruses, which may explain the lack of effect from vemurafenib (36, 56). In addition, it has been shown that the flavivirus West Nile virus requires lipid synthesis by the host but is independent of PI4P lipids and PI4KB (57). Whether West Nile virus or other positive RNA virus infections can be inhibited with vemurafenib needs to be studied in the future.

Finally, we were able to show that in addition to acute infection, vemurafenib was also able to eradicate persistent CVB4 infection. This is an important aspect since immunocompromised individuals can suffer from persistent enterovirus infections, which are difficult to treat with currently available drugs. Our findings open new possibilities for developing drugs against enteroviruses and evaluating further the potential of vemurafenib as a repurposed drug in clinical care.

## MATERIALS AND METHODS

**Cells, viruses, and reagents.** Adenocarcinomic human alveolar basal epithelial cells (A549; ATCC) were cultured in Dulbecco's modified eagle medium (DMEM; Gibco) supplemented with 10% fetal bovine serum (FBS; Gibco), 1% GlutaMAX (Gibco), and 1% penicillin and streptomycin antibiotics (Gibco). Min-6 cells (kindly provided by A. Abderrahmani, Lille, France), a pancreatic beta cell line, were cultured in DMEM medium (Gibco/BRL, Invitrogen, Gaithersburg, MD, USA) supplemented with 15% fetal calf serum (FCS; Sigma, St. Louis, MO, USA), 1% L-glutamine (Gibco/BRL), 50 $\mu$g/mL streptomycin, and 50 IU/mL penicillin (BioWhittaker). The human pancreatic ductal cell line Panc-1 (ATCC CRL-1469) was cultured in DMEM with 4.5 g/L glucose (Invitrogen, France) supplemented with 10% FCS, 1% L-glutamine, and 1% penicillin and streptomycin (Gibco). HeLa R19 cells were cultured at 37°C in 5% $CO_2$ in DMEM (Capricorn Scientific) supplemented with 10% FBS (Lonza) and 1% penicillin and streptomycin antibiotics.

Echovirus 1 (EV1; Farouk strain; ATCC), coxsackie virus A9 (CVA9; Griggs strain; ATCC), and human parechovirus 1 (HPEV1; Harris strain; a kind gift from Glyn Stanway, University of Essex, UK) were produced and purified as described (15) and isolated using sucrose gradients. Isolation of EV1 particles was carried out using a 10-mL linear 10 to 40% sucrose gradient, while for CVA9 and HPEV1, a 10-mL linear 5 to 20% sucrose gradient was used. The CV-B4 E2 diabetogenic strain (provided by Ji-Won Yoon, Julia McFarlane Diabetes Research Center, Calgary, AB, Canada) was grown in HEp-2 cells (BioWhittaker) in Eagle's minimum essential medium (MEM; Gibco) supplemented with 10% FCS, 1% L-glutamine, and 1% antibiotic (50 $\mu$g/mL streptomycin and 50 IU/mL penicillin). Supernatants were collected and then clarified at 2,500 rpm for 10 min. Virus titers were determined by limiting dilution assays for 50% tissue culture infectious doses ($TCID_{50}$) by the method of Reed-Muench on HEp-2 cells. Aliquots of virus preparations were then stored frozen at $-80$°C. RLuc-CVB3 viruses (strain Nancy) were obtained by transfecting HEK293T cells with RNA transcripts from the full-length infectious clone. CVB serotypes 1 to 6 were ATCC prototype strains (58). Poliovirus type 3 was a vaccine strain with a limited passage history obtained from the Finnish Institute for Health and Welfare (THL). EV71-VP1$_{97R167G}$ (genogroup C1) (59) was a kind gift from Caroline Tapparel (University of Geneva, Switzerland). Human rhinovirus B14 (HRVB14) was a kind gift from Timo Hyypiä (University of Helsinki). The GFP-expressing variant of adenovirus 5 was a kind gift from Vesa Turkki (Finvector, Kuopio, Finland).

The drug screening was carried out using the Institute of Molecular Medicine Finland (FIMM) oncology collection (FO4A) (https://www.helsinki.fi/en/infrastructures/drug-discovery-chemical-biology-and-screening/infrastructures/high-throughput-biomedicine/chemical-compound-libraries) consisting of approved and emerging investigational oncology drugs in 5 concentrations of each. In total, 462 drugs were tested.

Vemurafenib and Salirasib were purchased from Selleckchem, while sorafenib was from LC Laboratories.

**IC$_{50}$ and CC$_{50}$ determination.** A549 cells were seeded on 96-well plates and cultivated in DMEM supplemented with 10% FBS, 1% GlutaMAX, and 1% penicillin and streptomycin antibiotics for 24 h at 37°C. Vemurafenib was added to cells at different final concentrations diluted in DMEM supplemented with 1% FBS and 1% GlutaMAX and incubated for 1 h at 37°C. Next, EV1 or CVA9 was added (multiplicity of infection [MOI] of 10), and infection was allowed to proceed for 18 h at 37°C. The next day, cells were washed with PBS, after which the noninfected cells were stained with a solution containing crystal violet (8.3 mM crystal violet, 45 mM $CaCl_2$, 10% ethanol, 18.5% formaline, and 35 mM Tris base). Excess crystal violet stain was washed extensively with water, after which the crystal violet-stained cells were lysed with a lysis buffer (47.5% EtOH, 35 mM sodium citrate, and 12.5 mM HCl). Finally, the absorbance of the crystal violet was measured at 570 nm using Victor 220 X4 2030 multilabel reader (Perkin Elmer). For CVB1 to -6 serotypes, the infection assay was carried out as earlier described (58). The data were first normalized against the cell control without virus and vemurafenib. Next, the IC$_{50}$ was determined using nonlinear regression analysis in Graph Pad Prism software (version 6.07). Interpolation was used to determine the 50% cell viability (IC$_{50}$).

The cytotoxicity of vemurafenib was determined by first seeding A549 cells on a 96-well plate and cultivating them in DMEM supplemented with 10% FBS, 1% GlutaMAX, and 1% penicillin and streptomycin antibiotics for 24 h at 37°C. Next, vemurafenib was added at different concentrations diluted in DMEM supplemented with 1% FBS and 1% GlutaMAX and incubated for 6.5 h or 24 h at 37°C. Finally, the cell viability was determined using CellTiter-Glo viability assay (CTG; Promega) according to the instructions of the manufacturer. Luminescence was read using Victor 220 X4 2030 multilabel reader.

The data were normalized against the cell control without vemurafenib. Next, the $CC_{50}$ was determined using nonlinear regression analysis in Graph Pad Prism software (version 6.07). Interpolation was used to determine the 50% cell viability ($CC_{50}$).

**Adenovirus infection assay.** A549 cells were plated 1 day previously (300,000 cells/well) in DMEM supplemented with 10% FBS, 1% GlutaMAX, and 1% penicillin and streptomycin. The GFP-expressing variant of adenovirus 5 (MOI of 30) was added to DMEM supplemented with 1% FBS and 1% GlutaMAX with or without 5 $\mu$M vemurafenib, and the cells were infected for 17 h at 37°C. The cells were trypsinized and centrifuged at 5,000 rpm for 5 mins to collect the cell pellet. Finally, the cells were fixed with 4% paraformaldehyde (PFA) for 30 min. The samples were analyzed using Guava easyCyte flow cytometer by first gating the cells using forward and side scatter to exclude cell debris. Next, a second gate was used to evaluate the GFP fluorescence, and the percentage of GFP-positive cells from the total cell number (5,000 cells) was determined.

**Rhinovirus infection assay.** HeLa cells (a kind gift from Marino Zerial, Max Planck Institute, Dresden, Germany) were plated 1 day previously in DMEM supplemented with 10% FBS, 1% GlutaMAX, and 1% penicillin and streptomycin. HRVB14 was added to DMEM supplemented with 1% FBS and 1% GlutaMAX on ice and incubated for 1 h before excess virus was washed away. Fresh DMEM supplemented with 1% FBS and 1% GlutaMAX was added to cells with or without 5 $\mu$M vemurafenib, and the infection was allowed to proceed at 34°C for 6 h. Finally, the cells were fixed with 4% PFA and immunolabeled against the dsRNA (J2; Scicons). After the excess primary antibody was washed, Alexa Fluor 488-conjugated secondary antibody (Molecular Probes, Invitrogen, USA) was used to detect the primary antibody. During secondary antibody washes, 4',6-diamidino-2-phenylindole (DAPI) was also included to label the nuclei. Finally, the cells were mounted into Mowiol containing 1,4-diazabicyclo[2.2.2]octane (DABCO). Imaging was carried out using an Olympus XI81 inverted microscope equipped with a FluoView 1000 laser scanning confocal system. Images were taken using a USPLSAPO 60× oil objective. BioImageXD software was used to count the cells via DAPI staining and to quantify the dsRNA signal. The threshold for dsRNA signal was determined using a cell control to exclude the antibody background.

**Semliki Forest, vesicular stomatitis, and respiratory syncytial virus assays.** The experiment was performed in 96-well, black, clear-bottomed view plates (PerkinElmer). The HeLa cells were seeded in the 96-well plates at a density of $10^4$ cells/well and grown overnight at 37°C in a 5% $CO_2$ incubator. The growth medium was removed from each well and supplemented with infection medium (MEM with 0.2% BSA and 2 mM L-glutamine) containing vemurafenib at 5-$\mu$M and 10-$\mu$M concentrations. Ninety microliters per well of infection medium containing vemurafenib was added to cells and incubated at 37°C in a 5% $CO_2$ incubator for 30 min before infection with Semliki Forest virus (SFV), vesicular stomatitis virus (VSV), and respiratory syncytial virus (RSV). These HeLa cells were infected with viruses (provided by Giuseppe Balistreri) SFV-ZsGreen (60) and VSV-GFP (61) at the equivalent MOI of 0.1 (as determined by standard plaque assay in BHK-21 cells) for 6 h and RSV-GFP (62) for 18 h at the equivalent MOI of 1 (determined in HeLa HEP2 cells by infection center assay). For each well, 10 $\mu$L/well of viruses was added on top of drug-containing medium. After 6 h (SFV and VSV) and 18 h (RSV), the media were removed from each well, and cells were washed three times with 1× PBS. The infected cells were fixed with 4% PFA for 20 min at room temperature (RT). The fixation solution was removed, and cells were washed three times with PBS. The nuclei of the cells were stained with the DNA dye Hoechst (Thermo Fisher; catalog no. 62249) used at 1 $\mu$g/mL.

CellInsight Imager (Thermo Fisher; CellInsight software version 1.6.2.4-1.00x, build 6390) was used for automated fluorescence imaging. A 10× lens objective was used for imaging a with 386-nm filter to visualize Hoechst-stained nuclei and 485-nm filter to visualize the ZsGreen GFP fluorescence from infected cells. For each well, 9 different fields were imaged from both fluorescent channels. For image analysis, the open-source software CellProfiler 2.2.0 (www.cellprofiler.org) was used. Cells were identified by the intensity of fluorescence of Hoechst-stained nuclei using the Otsu algorithm built into the software. To differentiate between infected and noninfected cells, the intensity of green fluorescence was analyzed in each nucleus area. Noninfected wells were used as a control. The threshold of GFP fluorescence set to identify infected cells was set such that the rate of false-positive cells in noninfected cells was <0.1%. The cells were considered infected when the median intensity for GFP in the nuclear area was higher than the assigned threshold.

**Acute CVB4 infection in the pancreatic beta cell line.** The effect of vemurafenib on the CVB4-induced CPE in a mouse pancreatic beta cell line was studied by first seeding the Min-6 cells in 96-well tissue culture plates ($10^4$ cells/well/100 $\mu$L) and incubating for 24 h at 37°C and 5% $CO_2$. The next day, vemurafenib was added at final concentrations of 0.25, 0.5, 1, 2.5, 5, and 7.5 $\mu$M and incubated for 1 h at 37°C, after which 10 $\mu$L of CVB4 E2 was added to obtain an MOI of 1. The plates were incubated at 37°C with a humidified atmosphere at 5% $CO_2$ and checked by light microscopy at 24, 48, and 72 h. The viability of the cells was assessed using crystal violet.

**Persistent CVB4 infection in the pancreatic ductal cell line.** A model of persistent infection in Panc-1 cells with CVB4 was obtained as described previously (63, 64). Persistently infected Panc-1 cells were cultured in 6-well plates and treated with 7.5 $\mu$M vemurafenib each for 2 to 3 days. The cells were subcultured once a week. Supernatants were collected before each treatment renewal and conserved at −80°C before titration. Supernatants were 10-fold diluted and added to HEp-2 cells for titration.

Cells were washed with PBS 1×, scraped, and collected in 350 $\mu$L of RLT Plus lysis buffer (Qiagen, Courtaboeuf, France). Total RNA was extracted using the AllPrep DNA/RNA minikit (Qiagen) according to the manufacturer's instructions. The purified RNA was quantified by reading absorbance at 260 nm and 280 nm with a $\mu$Drop plate and spectrophotometer (Thermo Fisher Scientific). The SuperScript one-step RT-PCR (Thermo Fisher Scientific) was used for the enteroviral RNA positive-strand retrotranscription and cDNA amplification steps on the Applied GeneAmp PCR system 2400 thermocycler (PerkinElmer, Villebon sur Yvette, France). Seminested PCR was conducted afterward to assess the presence of low

quantities of viral RNA. The $\beta$-actin gene was coamplified as a control. Primers used for the enterovirus genome and $\beta$-actin gene and RT-PCR cycling conditions were previously described (65). Electrophoresis of the amplification products was then performed in 2% agarose gel containing 0.5 $\mu$g/mL ethidium bromide (Sigma-Aldrich) and visualized using the Gel Doc 2000 system (Bio-Rad, Marnes-la-Coquette, France).

**Mouse experiments.** Hsd:ICR(CD-1) female mice were provided by Envigo (Gannat, France) and handled in specific-pathogen-free conditions according to the guidelines of the 2010 EU directive. Experiments on mice were approved by the local Ethical Committee for Animal Experimentation (C2EA-75; Nord-Pas-de-Calais, France). Mice at the age of 3 weeks were inoculated intraperitoneally with vemurafenib dissolved in DMSO and diluted in PBS (10 mg/kg) or with DMSO diluted in PBS once a day (starting on day 1) for 5 days. The animals were inoculated intraperitoneally with CV-B4 E2 (6 $\times$ 10$^6$ TCID$_{50}$ in 200 $\mu$L PBS) on day 2.

The animals were sacrificed on day 6, blood was collected, and portions of each organ (pancreas and heart) were fixed in 10% formalin for histology or frozen for determination of viral titer. Frozen organs were weighed, crushed using a TissueRuptor (Qiagen, France), homogenized in 0.5 mL of PBS, and then centrifuged at 2,000 $\times$ $g$ for 10 min at 4°C. The supernatants were harvested to measure the titer of infectious particles (on HEp-2 cells), and titers were normalized to tissue weight. The results were expressed as log TCID$_{50}$ per gram. The limit of detection of the test was 0.75 log TCID$_{50}$/g.

**Immunofluorescence microscopy.** A549 cells were cultured on coverslips until subconfluence. Echovirus 1 or HPEV1 (2 $\times$ 10$^9$ PFU/mL) was added to cells in DMEM supplemented with 1% FBS and 1% GlutaMAX and incubated on ice for 1 h. Excess virus was washed with 0.5% BSA in PBS, after which the infection was allowed to proceed at 37°C in DMEM supplemented with 5% FBS and 1% GlutaMAX with or without 5 $\mu$M vemurafenib. In the time series experiment, vemurafenib was added to cells at the indicated time points postinfection at a final concentration of 5 $\mu$M, and the infection was followed until 6 hpi. Quantification of dsRNA structures was carried out at different time points postinfection (3, 4, and 6 h) with or without addition of vemurafenib on cells at 0 hpi at a final concentration of 5 $\mu$M.

In the ERK signaling experiment, A549 cells were plated on coverslips in DMEM supplemented with 10% FBS, 1% GlutaMAX, and 1% antibiotics. After 7 h, when the cells were attached to the coverslips, the medium was changed into DMEM supplemented with 0.5% FBS. The next day, cells were treated (or not, as a control) with 5 $\mu$M vemurafenib in DMEM supplemented with 0.5% FBS for 30 min at 37°C. Then, EV1 was added to cells (without removing the drug), and infection was allowed to proceed at 37°C for 2 h.

At the end of each assay, the cells were fixed with 4% PFA for 30 min at RT. Cells were permeabilized with 0.2% Triton X-100 and incubated with anti-EV1 antibody (66), anti-HPEV1 antibody (a kind gift from Timo Hyypiä, University of Turku, Turku, Finland), anti-dsRNA antibody (J2), or anti ERK1/2 antibody (Santa Cruz), washed, and incubated with Alexa Fluor555- or 488-conjugated secondary antibody (Molecular Probes, Invitrogen, USA). During secondary antibody washes, DAPI was also included to label the nuclei. Finally, the cells were mounted into Mowiol containing DABCO.

Imaging was carried out using Olympus Xl81 inverted microscope equipped with a FluoView 1000 laser scanning confocal system. Images were taken using a USPLSAPO 60$\times$ oil objective. Cells were counted via DAPI staining using BioImage XD software. In the time series and HPEV1 experiment, infected cells were counted manually. Quantification of dsRNA signal was carried out using BioImage XD, and the signal threshold was adjusted according to the mock infection control to exclude the antibody background.

**Sucrose gradients.** EV1 was metabolically labeled using [$^{35}$S]methionine and purified as described earlier (15). The virus, diluted in DMEM supplemented with 1% FBS and 1% GlutaMAX, was added to confluent A549 cells and incubated on ice for 30 min to synchronize the start of the infection. Next, cells were moved to 37°C with or without 5 $\mu$M vemurafenib. and infection was allowed to proceed for 4 h, after which the medium was removed, and cells were washed with PBS. To lyse the cells, 100 mM octyl-glucopyranoside diluted in 2 mM MgCl$_2$ in PBS was added to cells and incubated for 30 min on ice. Finally, the cell lysates were added on top of 10-mL linear 5 to 20% sucrose gradients made in R buffer (10 mM Tris-HCl [pH 7.5], 200 mM NaCl, 50 mM MgCl$_2$, and 10% [wt/vol] glycerol). The gradients were ultracentrifuged with an SW-41 rotor (35,000 rpm for 2 h) and fractioned into 500-$\mu$L aliquots from the top. The radioactivity of each fraction was quantified by a liquid scintillation counter (PerkinElmer) as counts per minute (cpm).

**RNA transfection.** EV1 RNA was isolated using High Pure viral RNA kit (Roche) according to the instructions of the manufacturer. A549 cells were cultivated on coverslips on 4-well plates until subconfluence. According to the manufacturer's instructions, 750 ng per well of viral RNA was transfected into the cells using Lipofectamine 3000. The cells were incubated with or without 5 $\mu$M vemurafenib in DMEM supplemented with 5% FBS and 1% GlutaMAX at 37°C for 6 h and were fixed with 4% PFA. After permeabilization with 0.2% Triton X-100, the cells were stained with anti-EV1 antibody (66), washed, and incubated with Alexa Fluor 555-conjugated secondary antibody (Molecular Probes, Invitrogen, USA). During secondary antibody washes, DAPI was also included to label the nuclei. Finally, the cells were mounted into Mowiol containing DABCO.

**Virus stability assay.** A real-time spectroscopic assay was carried out using Victor X4 2030 multilabel reader (PerkinElmer) with 485-nm and 535-nm excitation and 313-nm emission filters, respectively. One microgram of EV1 was mixed with 10$\times$ SYBR green II (Invitrogen) in either stabilizing buffer (2 mM MgCl$_2$ in PBS) or opening buffer (20 mM NaCl, 6 mM KH$_2$PO$_4$, 12 mM K$_2$HPO$_4$, and 0.01% faf-BSA) with or without 5 $\mu$M vemurafenib. In RNase controls, RNase (10 mg/mL, Thermo Fisher Scientific; catalog no. EN0531) was also added to the reaction mixture. The fluorescence was recorded at 37°C at 1-min intervals for 180 min in total.

**Viral protease assay.** Purified viral proteases 2A and 3C were expressed in *Escherichia coli* as described before (67). Viral proteases 2A and 3C of CVB3 were incubated with A549 cell homogenate, which was prepared as described earlier (68). In the reaction, 75 $\mu$g of the homogenate and 375 ng of 2A or 3C were incubated in buffer (20 mM HEPES [pH 7.4], 120 mM KCH$_3$COO, 4 mM Mg (CH$_3$COO)$_2$, and

5 mM dithiothreitol [DTT]) for 18 h at RT. Finally, a 4× SDS-PAGE sample buffer with mercaptoethanol was added as 1× final concentration to terminate the reaction. Next, samples were boiled and run in 4 to 20% MiniProtean TGX gradient gel (Bio-Rad) following a transfer to polyvinylidene difluoride (PVDF) membrane (Immobilon-P; Merck Millipore). The blots were blocked overnight with 5% BSA in 0.05% Tween in Tris-buffered saline (TBS) and immunolabeled with mouse anti-PABP (Santa Cruz) or mouse anti-G3BP1 (Santa Cruz). The primary antibodies were detected with corresponding horseradish peroxidase-conjugated secondary antibodies (Cell Signaling Technologies). Finally, chemiluminescent substrate (SuperSignal West Pico Plus; Thermo Fisher Scientific, Waltham, MA, USA) was incubated for 5 min. and the chemiluminescence was detected using ChemiDoc MP (Bio-Rad Laboratories Inc.). The total protein amount was revealed using a stain-free method by UV-activating the gel with ChemiDoc MP (Bio-Rad Laboratories Inc.) before transfer to PVDF membrane.

**Polymerase initiation and elongation assays.** CVB3 polymerase was expressed in and purified from *E. coli* as previously described (69). Rates of elongation complex formation and nucleotide incorporation were determined using two-hairpin primer-template RNAs labeled with a LI-COR Biosciences IRDye 800 at a modified thymidine in the hairpin loop sequence. CVB3 polymerase was incubated at room temperature with RNA1 and either 2.5% DMSO, 10 $\mu$M vemurafenib, or 100 $\mu$M vemurafenib for 15 min prior to addition of initiating NTPs in a buffer containing 75 mM NaCl, 50 mM HEPES, pH 6.5, 4 mM $MgCl_2$, and 2 mM Tris(2-carboxyethyl)phosphine hydrochloride (TCEP). Final concentrations of $3D^{pol}$ and RNA were 10 $\mu$M and 1 $\mu$M, respectively. After the incubation period, 1 $\mu$M RNA2 and 40 $\mu$M ATP and GTP were added to allow for incorporation of the initiating nucleotides. Time points were quenched at indicated times with 95% formamide, 50 mM EDTA, 75 mM NaCl, 0.01% bromophenol blue, and 0.02% xylene cyanol and PAGE resolved (20% acrylamide, 7 M urea). The gels were imaged on a LI-COR Biosciences Odyssey imager and quantified with ImageStudio (LI-COR Biosciences). Stopped-flow elongation assays were performed using a BioLogic SFM-4000 titrating stopped flow as previously described (69). CV3B elongation complexes were assembled by incubating 10 $\mu$M polymerase with 5 $\mu$M fluorescein-labeled RNA in 75 mM NaCl, 4 mM $MgCl_2$, 25 mM HEPES, pH 6.5, 2 mM TCEP, and 40 $\mu$M ATP for 15 min as previously described (69). Complexes were then diluted 200-fold and mixed with indicated amounts of vemurafenib or DMSO and 40 $\mu$M each NTP in the stopped flow at 22.5°C, pH 6.5. Elongation rates were calculated from the fluorescence data using KaleidaGraph (Synergy Software, Reading, PA) as described earlier (18).

**PI4K$\beta$ and OSBP immunofluorescence assay.** HeLa R19 cells grown to subconfluence on coverslips in 24-well plates were transfected with 300 ng of either FAPP1-PH-GFP (19) or pEGFP-hOSBP (20) plasmid using FuGene 6 (Promega) according to the manufacturer's protocol. After overnight expression, the cells were treated with a concentration range of vemurafenib for 1 h, fixed with 4% PFA in PBS for 15 min, and permeabilized with 0.5% Triton X-100 in PBS for 15 min. Subsequently, the cells were stained with anti-PI4K$\beta$ antibody (Merck) for 45 min, washed, and incubated with Alexa Fluor 568-conjugated secondary antibody (Invitrogen) and DAPI (Sigma) for 45 min. Cells were mounted on coverslides with Prolong Diamond antifade mountant (Invitrogen). The slides were imaged with a Leica SPE2 confocal microscope.

**Renilla luciferase assay.** HeLa R19 cells were grown to subconfluence in 96-well plates and infected with either RLuc-CVB3 wild type (WT), RLuc-CVB3-2C-AVIVAV, or RLuc-CVB3-3A-H57Y at an MOI of 0.1 at 37°C for 30 min. Next, the supernatant was removed, and compound containing DMEM was added to the cells. Cells were lysed at 7 hpi, and luciferase activity was determined using the *Renilla* luciferase assay system (Promega). Cell viability was determined using the CellTiter 96 AQueous One Solution cell proliferation assay (Promega). The $IC_{50}$ was calculated with nonlinear regressing using GraphPad Prism version 9.

## SUPPLEMENTAL MATERIAL

Supplemental material is available online only.

**SUPPLEMENTAL FILE 1**, DOCX file, 0.6 MB.

## ACKNOWLEDGMENTS

We acknowledge the Institute of Molecular Medicine Finland (FIMM) and Denis Kainov for helping in the initial drug screening.

This work was supported by FiDiPro and the Jane & Aatos and Erkko Foundation (M.L. and V.M.). Parts of this project were supported by the University of Helsinki Doctoral Program in Microbiology and Biotechnology (R.O.) and the Academy of Finland Research Grant 318434 (G.B. and R.O.) and NIH grant R01 AI059130 (O.P.). High-throughput imaging was performed at the Light Microscopy Unit of the University of Helsinki.

H.H. is a minor shareholder and member of the board of Vactech Ltd., which develops vaccines against picornaviruses. No other potential conflicts of interest relevant to this article were reported.

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
