## [Reviewer comments · Microbiology Spectrum]

Microbiology Spectrum

Vemurafenib inhibits acute and chronic enterovirus infection by affecting cellular kinase phosphatidylinositol 4-kinase type III β

Mira Laajala, Marleen Zwaagstra, Mari Martikainen, Magloire Nekoua, Mehdi Benkahla, Famara Sane, Emily Gervais, Grace Campagnola, Anni Honkimaa, Amir-Babak Sioofy Khojine, Heikki Hyöty, Ravi Ojha, Marie Bailliot, Giuseppe Balistreri, Olve Peersen, Didier Hober, Frank van Kuppeveld, and Varpu Marjomäki

Corresponding Author(s): Varpu Marjomäki, Jyväskylän yliopisto

Review Timeline:

Submission Date:	February 6, 2023
Editorial Decision:	March 20, 2023
Revision Received:	June 14, 2023
Accepted:	June 14, 2023

Editor: Ralph Tripp

Reviewer(s): Disclosure of reviewer identity is with reference to reviewer comments included in decision letter(s). The following individuals involved in review of your submission have agreed to reveal their identity: Minetaro Arita (Reviewer #2)

Transaction Report:

DOI: <https://doi.org/10.1128/spectrum.00552-23>

March 20, 2023

Prof. Varpu Marjomäki
Jyvaskylan yliopisto
Biological and Environmental Sciences / Nanoscience center
Survontie 9
Jyväskylä, Jyväskylä 40500
Finland

Re: Spectrum00552-23 (Vemurafenib inhibits acute and chronic enterovirus infection by affecting cellular kinase phosphatidylinositol 4-kinase type III β)

Dear Prof. Varpu Marjomäki:

please address the reviewer's concerns

Link Not Available

Sincerely,

Ralph Tripp

Journals Department
Reviewer comments:

Reviewer #1 (Comments for the Author):

The manuscript by Laajala et al., examined the antiviral activity of vemurafenib against enteroviruses and several other DNA and RNA viruses. Specifically, they showed that Vemurafenib was effective against group B and C enteroviruses as well as rhinovirus but not parechovirus, Semliki Forest virus, adenovirus, or human respiratory syncytial virus. They also tried to understand the mechanism of action of this antiviral drug. They found that the antiviral effect of vemurafenib was due to the inhibitory of cellular PI4KB, but not RAF/MEK/ERK pathway. Overall, the study was well designed, the procedures were well described, and the results are interesting. The manuscript was also well-written and can be potentially published in this journal.

However, several points should be addressed. These comments are:

1. When talking about antiviral drugs, the readers always want to know the IC50 and CC50 of the drug. The authors should calculate the IC50 of vemurafenib against selected enteroviruses.
2. Fig.2C, RT-PCR data, please show the DNA bands in agarose gel.
3. Fig.3B, error bar was not shown?
4. Fig.5A, please indicate the molecular weight of the targeted proteins.
5. Fig.6, can the authors show the Western blot analysis of PI4KB in the increasing dose of vemurafenib? This is the most important data on mechanistic study.

Reviewer #2 (Comments for the Author):

Laajala et al. reported an FDA-approved RAF kinase inhibitor vemurafenib potentially targets cellular PI4KB for anti-EV activity. Anti-EV-A71 activity of vemurafenib has been reported by Hu et al. in 2022. The potential importance of the current work might be in the target of vemurafenib for the anti-EV activity.

Some concerns on technical issues and the writing are as below.

Major points:

1. I could not agree that too many important references related to PI4KB and OSBP were missed in an awkward way. I would strongly suggest the authors completely correct this issue for the readership of this manuscript.
2. L35, Fig.1: Values of CC50(after short(7 h) and long (equal or more than 2 days) treatment), EC50, and selectivity index of vemurafenib for virus infection and of EC50 for killing of melanoma cells and normal cells should be provided and discussed on the potential potency as antiviral. Hu et al. already reported that the EC50 value of vemurafenib to EV-A71 infection was at the order of nM (Hu et al., 2022, Pharmaceuticals).
3. The labels in Fig.1C should be magnified. The current figure is invalid.
4. IC50 values of vemurafenib for BRAF (V600E/K) and PI4KB activities should be provided and discussed on the potential potency as above. Even if the inhibitory effect of vemurafenib on PI4KB activity could not be evaluated for any reason, this important point should be clarified as a limitation of this study to avoid misleading the readers. Most of the current discussion is based on a possible inference that vemurafenib is a PI4KB inhibitor.
5. L141, 152, L237: Conditions of viral infection are not clearly described; at least MOI should be described for all the viruses.
6. L128: An MOI of 400 seemed too high for evaluation of the antiviral effect against EV infection. An explanation of the condition may be included.
7. Fig.2C. Explanation of the gradual increment of the viral titer in Mock-treated cells should be added.
8. L356: Information on used drug library or the compounds should be provided.
9. L606: A c-RAF kinase inhibitor (GW5074) was known as a PI4KB inhibitor (Arita et al., 2008, JGV, 2011, JVI). This fact should be included here because this working hypothesis was not a novel one.
10. Fig.6A: Intracellular localization of PI4KB in OSW-1 and ITZ-treated cells was almost invisible. The current data set is not sufficient to show the effect of vemurafenib on the relocalization of PI4KB. Clear data would be required.
11. Fig.6B: The effect of ITZ needs to be clarified.
12. Fig.6C: Observed resistance was partial, and possibly affected by the high cytotoxicity of vemurafenib. The statistical significance should be shown to support the resistance of the CVB3(H57Y) mutant. In addition, the cytotoxicity should be clearly discussed with the CC50 values in the used cells.
13. Fig. S6C. CC50 and EC50 values should be shown.
14. L724-740: The target population of the anti-EV drug, cytotoxicity in vitro, and side effects in vivo of vemurafenib should be clearly discussed in the context of the drug reposition. Vemurafenib is not applicable to young healthy children.
15. L746: Complete resistance of a poliovirus mutant against PI4KB/OSBP inhibitors has been reported (Arita and Bigay, 2019, ACS Infect Dis).
16. L717: The link between GBF1 and PI4KB accumulation has not been clarified to date.
17. L719: Functional link between PI4KB and OSBP was suggested in a trilogy of papers (Arita et al., 2011 JVI, 2013 JVI, 2014, Microbiol Immunol). The potential role of OSBP should be discussed based on these refs.

Minor points:

1. L79 and 704: The primary target of PI4KB is considered viral 3AB rather than 3A (Arita, 2016, ACS Infect Dis, Melita et al., 2017, Cell Rep).
2. Information on all the used viruses in the current study should be provided in the Cells and Virus section in Materials and methods with Genbank accession number with the information of the source (e.g., ATCC).
3. L609. An original reference for PI4KB relocalization by the inhibitors should be included (van der Schaar et al., 2012, Cell Res).
4. L618. Original references for OSBP relocalization by the inhibitors should be included (Ridgway et al., 1992, JCB, Arita et al., 2013, JVI).

Staff Comments:

Preparing Revision Guidelines

Please return the manuscript within 60 days; if you cannot complete the modification within this time period, please contact me. If you do not wish to modify the manuscript and prefer to submit it to another journal, please notify me of your decision immediately so that the manuscript may be formally withdrawn from consideration by Microbiology Spectrum.

Point by point responses

Reviewer comments:

Reviewer #1 (Comments for the Author):

The manuscript by Laajala et al., examined the antiviral activity of vemurafenib against enteroviruses and several other DNA and RNA viruses. Specifically, they showed that Vemurafenib was effective against group B and C enteroviruses as well as rhinovirus but not parechovirus, Semliki Forest virus, adenovirus, or human respiratory syncytial virus. They also tried to understand the mechanism of action of this antiviral drug. They found that the antiviral effect of vemurafenib was due to the inhibitory of cellular PI4KB, but not RAF/MEK/ERK pathway. Overall, the study was well designed, the procedures were well described, and the results are interesting. The manuscript was also well-written and can be potentially published in this journal. However, several points should be addressed. These comments are:

1. When talking about antiviral drugs, the readers always want to know the IC50 and CC50 of the drug. The authors should calculate the IC50 of vemurafenib against selected enteroviruses.

We thank the reviewer for this suggestion. We have now added the IC50 values against selected enteroviruses (Figure 1A) and CC50 (Figure 1B).

2. Fig.2C, RT-PCR data, please show the DNA bands in agarose gel.

We thank the reviewer for this suggestion. The gel image has now been added to Figure 2C.

3. Fig.3B, error bar was not shown?

We thank the reviewer for this notion. The error bars were accidentally left out from the graph, but they have now been added in Figure 3B.

4. Fig.5A, please indicate the molecular weight of the targeted proteins.

We have now added the molecular weights in Figure 5A.

5. Fig.6, can the authors show the Western blot analysis of PI4KB in the increasing dose of vemurafenib? This is the most important data on mechanistic study.

We thank the reviewer for this comment. We think that the western blot may not show huge difference between control and vemurafenib treated cells. Instead, the difference in PI4KB localization is very clearly visible and can be seen in Figure 6A. This translocation has also been shown with other PI4KB inhibitors earlier (Van Der Schaar et al. 2012). We have now also added this reference in the results section, page 21.

Reviewer #2 (Comments for the Author):

Laajala et al. reported an FDA-approved RAF kinase inhibitor vemurafenib potentially targets cellular PI4KB for anti-EV activity. Anti-EV-A71 activity of vemurafenib has been reported by Hu et al. in 2022. The potential importance of the current work might be in the target of vemurafenib for the anti-EV activity.

Some concerns on technical issues and the writing are as below.

Major points:

1. I could not agree that too many important references related to PI4KB and OSBP were missed in an awkward way. I would strongly suggest the authors completely correct this issue for the readership of this manuscript.

We thank the reviewer for this comment and apologize that there were not enough references concerning PI4KB and OSBP. We have now added many new references both in the results and discussion sections. Please see our more detailed comments below.

2. L35, Fig.1: Values of CC50(after short (7 h) and long (equal or more than 2 days) treatment), EC50, and selectivity index of vemurafenib for virus infection and of EC50 for killing of melanoma cells and normal cells should be provided and discussed on the potential potency as antiviral. Hu et al. already reported that the EC50 value of vemurafenib to EV-A71 infection was at the order of nM (Hu et al., 2022, Pharmaceuticals).

We thank the reviewer for this suggestion. We have now added the IC50 values of selected enteroviruses (Figure 1A) and CC50 data of vemurafenib for short and long timepoints (Figure 1 B) as well as SI (Figure 1B) in A549 cells. We have not done our studies on melanoma or normal cells as the idea was to study the effects in A549 cells lacking the V600 mutation. See also page 16.

3. The labels in Fig.1C should be magnified. The current figure is invalid.

We apologize that the labels in Fig.1C were not clearly visible. The figure has now been corrected.

4. IC50 values of vemuferanib for BRAF (V600E/K) and PI4KB activities should be provided and discussed on the potential potency as above. Even if the inhibitory effect of vemuferanib on PI4KB activity could not be evaluated for any reason, this important point should be clarified as a limitation of this study to avoid misleading the readers. Most of the current discussion is based on a possible inference that vemuferanib is a PI4KB inhibitor.

We thank the referee for the great suggestion. We were able to perform the activity assay on BRAF (V600E/K). This result is shown here below for the referee to see. There was only one vendor, Abcam, who originally suggested that they could provide PI4KB for activity testing. However, Abcam ran into trouble trying to make the product, and they continuously postponed the shipment, and, after waiting for several weeks, Abcam again notified about yet another long waiting time. Therefore, we decided to submit the revision now as there was no clear indication that we could get any product in the end. We are sorry for this. Therefore, we have added this as the limitation of the study as the referee suggested. Also, we feel that adding the BRAF activity alone is not needed information as we lack the PI4KB data. Please, see page 23-24.

5. L141, 152, L237: Conditions of viral infection are not clearly described; at least MOI should be described for all the viruses.

We thank the reviewer for this comment. In the rhinovirus, EV1 and HPEV1 assays we have used ice binding where the virus is attached to the receptor through 1h ice binding after which excess virus is thoroughly washed. Hence, determination of exact MOI is not possible, but instead, only the receptor bound virus is allowed to infect the cells.

In adenovirus assay, we have used MOI 30 and the info has been added in the methods section, page 7.

6. L128: An MOI of 400 seemed too high for evaluation of the antiviral effect against EV infection. An explanation of the condition may be included.

The reviewer is right, very high MOI was used here but still vemurafenib showed efficacy against the infection. However, as we have added the IC50 data of EV1 and CVA9 and all the CVB serotypes in Figure 1A, we have now removed this data.

7. Fig.2C. Explanation of the gradual increment of the viral titer in Mock-treated cells should be added.

We thank the reviewer for this notion. The persistence of CVB4 in human Panc-1 cells has been previously reported as "carrier state" persistence characterized by a productive infection of a part of cells in the culture since the proportion of VP1-positive cells ranges from about 1 to 5% (Sane et al., 2013).

The gradual increment of the viral titer in mock-treated cells can be explained by the gradual increase in the number of productively infected cells throughout the follow-up of the cultures.

Reference:

Sane F, Caloone D, Gmyr V, Engelmann I, Belaich S, Kerr-Conte J, Pattou F, Desaillood R, Hober D. Cocksackievirus B4 can infect human pancreas ductal cells and persist in ductal-like cell cultures which results in inhibition of Pdx1 expression and disturbed formation of islet-like cell aggregates. Cell Mol Life Sci. 2013 Nov;70(21):4169-80. doi: 10.1007/s00018-013-1383-4.

8. L356: Information on used drug library or the compounds should be provided.

We thank the reviewer for this comment. We have now added info in the materials and methods section about the drug library and the companies where vemurafenib, salirasib and sorafenib were purchased, page 6.

9. L606: A c-RAF kinase inhibitor (GW5074) was known as a PI4KB inhibitor (Arita et al., 2008, JGV, 2011, JVI). This fact should be included here because this working hypothesis was not a novel one.

We thank the reviewer for this comment and for pointing Arita et al. 2008 reference. We were unaware that GW5074 is originally a RAF inhibitor. Interestingly, Arita et al. were not able to grow escape mutants in the presence of this RAF inhibitor and we noticed the same thing with vemurafenib. By using a method described in many papers (refs 1,2 and 3 below), escape mutants did not develop during 3 days of CVB3 infection (MOI 10) in the presence of 5 µM vemurafenib in two repeats with 144 sample wells in each, distributed on three 96-well plates (see a representative figure below). The cell viability was determined

using crystal violet staining and absorbance measurement of the stain. We have now added Arita et al. 2008 reference in the discussion section and more discussion on escape mutants. page 25.

References:

1. Leen Delang, Nidya Segura Guerrero, Ali Tas, Gilles Qu rat, Boris Pastorino, Mathy Froeyen, Kai Dallmeier, Dirk Jochmans, Piet Herdewijn, Felio Bello, Eric J. Snijder, Xavier de Lamballerie, Byron Martina, Johan Neyts, Martijn J. van Hemert, Pieter Leyssen, Mutations in the chikungunya virus non-structural proteins cause resistance to favipiravir (T-705), a broad-spectrum antiviral, *Journal of Antimicrobial Chemotherapy*, Volume 69, Issue 10, October 2014, Pages 2770–2784, <https://doi.org/10.1093/jac/dku209>
2. Bauer L, Manganaro R, Zonsics B, Hurdiss DL, Zwaagstra M, Donselaar T, Welter NGE, van Kleef RGD, Lopez ML, Bevilacqua F, Raman T, Ferla S, Bassetto M, Neyts J, Strating JRPM, Westerink RHS, Brancale A, van Kuppeveld FJM. Rational design of highly potent broad-spectrum enterovirus inhibitors targeting the nonstructural protein 2C. *PLoS Biol.* 2020 Nov 6;18(11):e3000904. doi: 10.1371/journal.pbio.3000904. PMID: 33156822; PMCID: PMC7673538.
3. Tijsma A, Thibaut HJ, Spieser SA, De Palma A, Koukni M, Rhoden E, Oberste S, P rstinger G, Volny-Luraghi A, Martin J, Marchand A, Chaltin P, Neyts J, Leyssen P. H1PVAT is a novel and potent early-stage inhibitor of poliovirus replication that targets VP1. *Antiviral Res.* 2014 Oct;110:1-9. doi: 10.1016/j.antiviral.2014.07.003. Epub 2014 Jul 17. PMID: 25043639.

10. Fig.6A: Intracellular localization of PI4KB in OSW-1 and ITZ-treated cells was almost invisible. The current data set is not sufficient to show the effect of vemurafenib on the relocalization of Pi4KB. Clear data would be required.

We have now increased the intensity of the figures in order to try to make the signal more visible. However, the fact is that for OSW-1 and ITZ treated cells, the signal is very dispersed as they do not inhibit PI4KB. When comparing ITZ and OSW-1 to 1 μ M vemurafenib (which also does not inhibit PI4KB) there is not a big difference in PI4KB signal. New versions of the figures have been added to the manuscript. See also here below.

11. Fig.6B: The effect of ITZ needs to be clarified.

For ITZ, there is a redistribution of OSBP to the Golgi, similar to OSW-1, but however, due to inhibition of OSBP, the signal is less pronounced than seen for OSW-1.

12. Fig.6C: Observed resistance was partial, and possibly affected by the high cytotoxicity of vemurafenib. The statistical significance should be shown to support the resistance of the CVB3(H57Y) mutant. In addition, the cytotoxicity should be clearly discussed with the CC50 values in the used cells.

As suggested by the referee, we have now calculated the values:

Between CVB3 WT and AVIVAV: $p = 0,09$

Between CVB3 WT and H57Y: $p = 0,0001$

These have been added to the figure legend.

13. Fig. S6C. CC50 and EC50 values should be shown.

The IC50 value is already in the figure. Concerning the CC50 value, 100 μM of vemurafenib results in 60% cell viability, so CC50 is probably around 100 μM

14. L724-740: The target population of the anti-EV drug, cytotoxicity in vitro, and side effects in vivo of vemurafenib should be clearly discussed in the context of the drug reposition. Vemurafenib is not applicable to young healthy children.

We thank the reviewer for suggesting more discussion about the target population and safety. We have added more text in pages 24-25.

15. L746: Complete resistance of a poliovirus mutant against PI4KB/OSBP inhibitors has been reported (Arita and Bigay, 2019, ACS Infect Dis).

We thank the reviewer for pointing this reference. It has now been added to the discussion section along with Van Der Schaar et al. 2012, page 25

See also our answer to point 9 above.

16. L717: The link between GBF1 and PI4KB accumulation has not been clarified to date.

We thank the reviewer for this comment. We have now added discussion related to this and clarified our point, page 24.

17. L719: Functional link between PI4KB and OSBP was suggested in a trilogy of papers (Arita et al., 2011 JVI, 2013 JVI, 2014, Microbiol Immunol). The potential role of OSBP should be discussed based on these refs.

We thank the reviewer for pointing out these references. They have now been added in the discussion section page 24.

Minor points:

1. L79 and 704: The primary target of PI4KB is considered viral 3AB rather than 3A (Arita, 2016, ACS Infect Dis, Melia et al., 2017, Cell Rep).

We thank the reviewer for pointing this out and providing the references. We have now edited the introduction and discussion sections and added these references. Pages 3 and 23.

2. Information on all the used viruses in the current study should be provided in the Cells and Virus section in Materials and methods with Genbank accession number with the information of the source (e.g., ATCC).

We have now added the missing information about the viruses in the methods section.

3. L609. An original reference for PI4KB relocalization by the inhibitors should be included (van der Schaar et al., 2012, Cell Res).

We thank the reviewer for this comment. We have now added this reference in the results section, page 21.

4. L618. Original references for OSBP relocalization by the inhibitors should be included (Ridgway et al., 1992, JCB, Arita et al., 2013, JVI).

We thank the reviewer for this comment. We have now added these references in the results section as well as Starting et al. 2015, Cell Rep, page 21.

June 14, 2023

Prof. Varpu Marjomäki
Jyvaskylan yliopisto
Biological and Environmental Sciences / Nanoscience center
Survontie 9
Jyväskylä, Jyväskylä 40500
Finland

Re: Spectrum00552-23R1 (Vemurafenib inhibits acute and chronic enterovirus infection by affecting cellular kinase phosphatidylinositol 4-kinase type III β)

Dear Prof. Varpu Marjomäki:

The revised manuscript is acceptable.

Your manuscript has been accepted, and I am forwarding it to the ASM Journals Department for publication. You will be notified when your proofs are ready to be viewed.

Sincerely,

Ralph Tripp
Editor, Microbiology Spectrum
